# Obesity-induced galectin-9 is a therapeutic target in B-cell acute lymphoblastic leukemia

Miyoung Lee[1], Jamie A. G. Hamilton[1], Ganesh R. Talekar[1], Anthony J. Ross[2], Langston Michael[3], Manali Rupji [4], Bhakti Dwivedi[4], Sunil S. Raikar [1], Jeremy Boss [5], Christopher D. Scharer [5], Douglas K. Graham[1], Deborah DeRyckere[1], Christopher C. Porter[1] & Curtis J. Henry [1✉]

The incidence of obesity is rising with greater than 40% of the world's population expected to be overweight or suffering from obesity by 2030. This is alarming because obesity increases mortality rates in patients with various cancer subtypes including leukemia. The survival differences between lean patients and patients with obesity are largely attributed to altered drug pharmacokinetics in patients receiving chemotherapy; whereas, the direct impact of an adipocyte-enriched microenvironment on cancer cells is rarely considered. Here we show that the adipocyte secretome upregulates the surface expression of Galectin-9 (GAL-9) on human B-acute lymphoblastic leukemia cells (B-ALL) which promotes chemoresistance. Antibody-mediated targeting of GAL-9 on B-ALL cells induces DNA damage, alters cell cycle progression, and promotes apoptosis in vitro and significantly extends the survival of obese but not lean mice with aggressive B-ALL. Our studies reveal that adipocyte-mediated upregulation of GAL-9 on B-ALL cells can be targeted with antibody-based therapies to overcome obesity-induced chemoresistance.

[1] Department of Pediatrics, Emory University School of Medicine and Aflac Cancer and Blood Disorders Center, Children's Healthcare of Atlanta, Atlanta, GA, USA. [2] Riley Pediatric Cancer and Blood Diseases, Riley Children's Health, Indiana University School of Medicine, Indianapolis, Indiana, IN, USA. [3] Wake Forest University, Winston-Salem, NC, USA. [4] Bioinformatics and Biostatistics Shared Resource, Winship Cancer Institute, Atlanta, GA, USA. [5] Department of Microbiology and Immunology, Emory University School of Medicine, Atlanta, GA, USA. ✉email: curtis.j.henry@emory.edu

O besity rates are reaching epidemic proportions with persons with obesity predicted to comprise a third of the world's population by 2030[1,2]. This alarming trend poses serious health concerns given the increased incidence of both solid and hematological malignancies associated with obesity[3–5]. Furthermore, obesity significantly increases mortality rates in patients with various malignancies[6–8].

The relationship between obesity and poor therapeutic outcomes has largely been attributed to the accumulation of adipocytes, which are thought to significantly alter drug pharmacokinetics;[9–11] however, this correlation is still under investigation[12]. We are just beginning to dissect and appreciate how an adipocyte-rich microenvironment impacts hematopoiesis and hematological malignancies. Recently, obesity has been shown to compromise hematopoiesis by reducing the function of hematopoietic stem cells (HSCs) resulting from increased oxidative stress and the upregulation of the transcription factor GFI1[13]. Although obesity compromises the function of non-malignant HSCs, the converse is true for leukemia-initiating cells (LICs). In recent studies, adipocytes supported the growth of LICs and protected them from the cytotoxic effects of chemotherapies by secreting lipids and amino acids that LICs can use for β-oxidation and glutaminolysis, respectively[14,15]. This is not a passive process given the recent findings that leukemia cells in gonadal adipose tissue induce lipolysis in adipocytes, and the secreted lipids are subsequently transported into leukemia cells using the fatty acid transporter CD36[16]. These results highlight the impact of adipocytes on the function of HSCs and LICs. Despite these discoveries, we still lack effective treatment strategies to improve therapeutic outcomes in overweight and patients with obesity[17–20]. Given the mounting evidence that adipocytes directly alter the function of leukemia cells, we sought to identify obesity-induced pathways in B-ALL cells that could be targeted alone or in combination with front-line chemotherapies to improve therapeutic outcomes.

In this work, we show that adipocytes induce Galectin-9 (GAL-9) surface expression on human B-ALL and that the induction of this lectin is protective against environmental and chemotherapy-induced cytotoxicity. We also report that GAL-9 surface expression is higher on B-ALL cells isolated from pediatric patients with obesity relative to lean patients. Furthermore, in relapsed disease, higher GAL-9 gene expression levels are associated with poor survival outcomes. Notably, we demonstrate that αGAL-9 antibody treatment is highly cytotoxic to B-ALL cells in vitro and significantly extends the survival of obese mice with B-ALL.

## Results

**Obesity potentiates B-ALL progression.** Leukemia is the most diagnosed cancer in children (20–25% of all diagnosed cases) and the 6th most diagnosed cancer in the United States each year[21–23]. Given the rising obesity rates in children and adults[24–26], along with the association between obesity and carcinogenesis[27–29], we sought to determine how obesity impacts B-ALL progression in mice. Two-month-old, C57BL/6 mice were fed control (10% fat) or high-fat (60% fat) diets for two months, leading to significant increases in the weight (Fig. 1a) of young mice fed high fat diets. Lean and obese mice were challenged with GFP-expressing BCR-ABL1+ Arf−/− murine B-ALL (mB-ALL) cells. In this model, B-cell leukemia can be established in immunocompetent mice without myeloablation, thus leaving the immune system and microenvironment unperturbed[30]. We observed dramatic differences in the survival of lean and obese mice challenged with mB-ALL. Specifically, 40% of lean mice survived challenge with mB-ALL, whereas, every obese mouse succumbed to leukemia-induced morbidity within one month of challenge (Fig. 1b). These data are consistent with previously

published findings demonstrating that obesity significantly increased mortality in mice challenged with B-ALL[15,31].

**Adipocytes alter the function of B-ALL cells.** To study the impact of adipocytes on B-ALL cells, we differentiated bone marrow stromal cells (BMSC) into adipocytes as previously described[32–34]. Using this method, adipocyte differentiation occurred rapidly and became pronounced by day 3 of culture, as indicated by the massive accumulation of lipid droplets (Supplementary Fig. 1a, b). Furthermore, adipocytes expressed higher levels of Fatty Acid Binding Protein 4 (FABP4), a marker of differentiated adipocytes[35], relative to parental BMSCs (Supplementary Fig. 1c). The secreted levels of proinflammatory cytokines (Supplementary Fig. 2a) and chemokines (Supplementary Fig. 2b) were also consistently higher for adipocytes relative to BMSC, including for IL-6 and TNF-α (Supplementary Fig. 2c), which are hallmarks of chronic inflammation[3,36–38].

After validating adipocyte differentiation, we next sought to determine how adipocyte-secreted factors impact the function of human B-ALL cells. One of the most striking phenotypes observed in adipocyte-conditioned medium (ACM)-exposed human B-ALL cells was the induction of extensive cellular aggregation which occurred in every human B-ALL cell line tested (representative data from 2 of 8 cell lines are shown; Fig. 2a); whereas, B-ALL cells remained in a single cell suspension when cultured in unconditioned medium (RPMI) or stromal cell-conditioned medium (SCM). Given these phenotypic changes, we next assessed the impact of unconditioned medium, SCM, or ACM on the function of human B-ALL cells. To this end, we cultured 5 human B-ALL cell lines in each condition for 3 days to assess the impact on proliferation, survival, and proteins regulating both processes. When B-ALL cells were cultured in unconditioned medium or SCM, the total number of leukemia cells increased by 6 to 10-fold over 3 days; whereas, all of the B-ALL cell lines cultured in ACM or recombinant TNF-α (a component of ACM) exhibited a 2 to 4-fold increase in density over the same period (Fig. 2b and Supplementary Fig. 1d). The lower number of human B-ALL cells observed in ACM cultures at day 3 could not be fully explained by increased cell death given that significant increases in apoptosis were observed in only 2 of the 5 human B-ALL cell lines cultured in ACM, and the extent of cell death never exceeded 30% over this period (Fig. 2c).

Given that increased apoptosis could not explain the reduced number of human B-ALL cells cultured in ACM relative to the other conditions tested, we hypothesized that adipocyte-secreted factors may induce cellular senescence in human B-ALL cells. To test this hypothesis, we cultured human B-ALL cells in unconditioned, SCM, or ACM for 24 h and measured β-Galactosidase (β-Gal) activity to determine the induction of senescence. Indeed, relative to responses observed in human B-ALL cells cultured in unconditioned medium and SCM, we observed significant increases in β-Gal activity in every human B-ALL cell line cultured in ACM (Fig. 2d). Furthermore, pathways involved in inducing senescence, including AKT[39–41], ERK[42–44], XIAP[45–47], and BCL-xL[48–50], were activated to varying degrees, in human B-ALL cells when cultured in ACM (Fig. 2e and Supplementary Fig. 3). Overall, our results demonstrate that adipocyte-secreted factors induce the aggregation of human B-ALL cells and induce cellular senescence, the latter of which we hypothesized would reduce chemosensitivity.

**Adipocytes promote chemoresistance.** Given these observations, we next sought to determine how pre-conditioning B-ALL cells with ACM impacted their responses to chemotherapy treatment. In patients with obesity, leukemia cells would be exposed to an

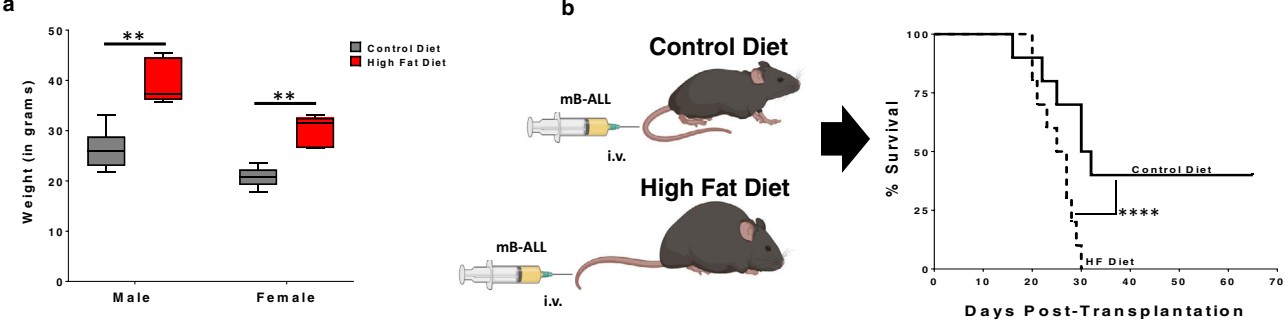

**Fig. 1 Obesity potentiates B-ALL progression. a** C57BL/6 mice were maintained on control (Ctrl; 10% fat) and high-fat (HF; 60% fat) diets for 2 months, leading to a significant increase in weight gain in mice fed high-fat diets. Box and Whisker plots display the median, 25th, and 75th percentile. The whiskers extend from the minima to the maxima, respectively. Significance is denoted by **$p < 0.01$ using a two-sided Student's $t$-test ($n = 10$ mice/group). **b** Mice fed control and high-fat diets were inoculated intravenously with $10^5$ Bcr-Abl$^+$ $Arf-/-$ murine B-ALL cells (mB-ALL) and survival was monitored. ****$P < 0.0001$, log-rank test, $n = 10$ mice/group.

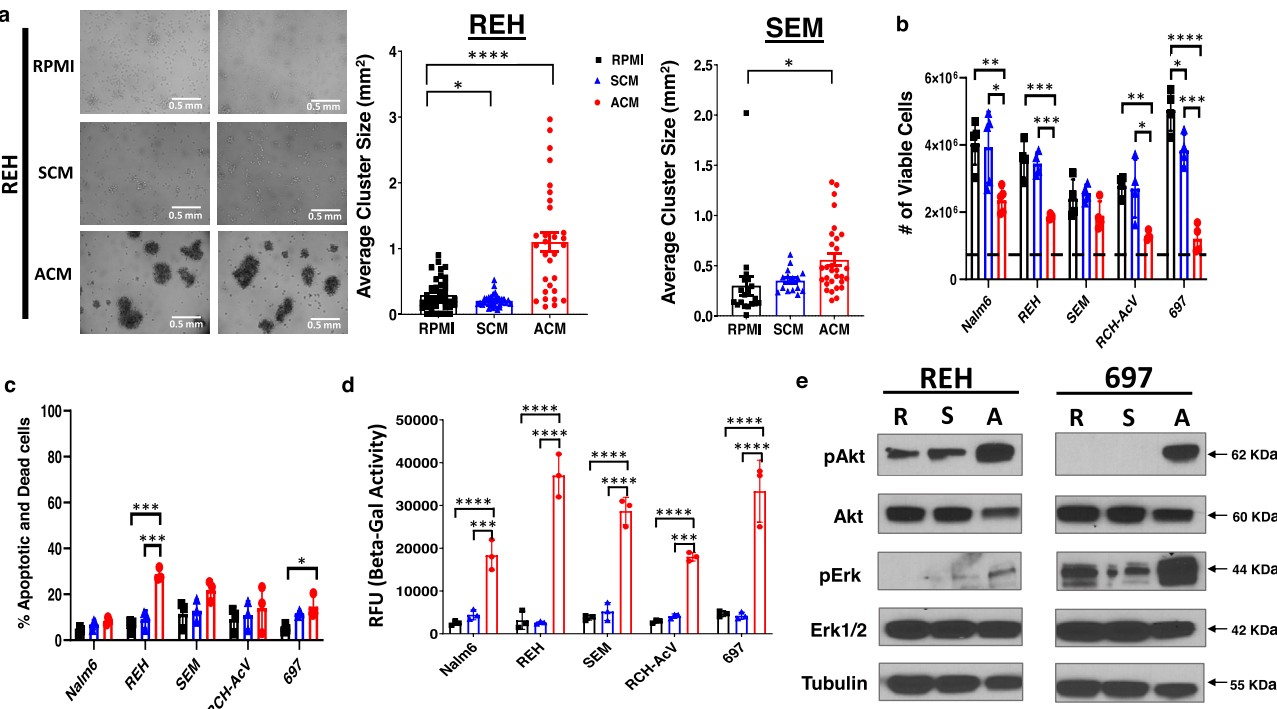

**Fig. 2 Adipocyte-secreted factors promote cellular aggregation and reduce the proliferation of human B-ALL cells. a** REH and SEM B-ALL cell lines were cultured in unconditioned medium (RPMI), SCM, or ACM. Brightfield microscopy images were taken after 3 days and cluster sizes were determined by ImageJ analysis. **b** Human B-ALL cell lines (Nalm6, REH, SEM, RCH-AcV, and 697) were plated in RPMI, SCM, or ACM at a starting density of $5 × 10^5$ cells/well (indicated by the dotted line). The number of viable cells was determined on day 3 of culture using trypan blue exclusion assays. **c** The percentage of apoptotic or dead cells in **b** was determined using Annexin V/PI assays and analyzed using flow cytometry. **d** Human B-ALL cell lines were cultured in RPMI, SCM, or ACM for 24 h. Lysates were harvested and β-Galactosidase activity was determined using the Senescence β-Galactosidase Activity Assay Kit (Cell Signaling Technology) per the manufacture's protocol. **e** Human B-ALL cell lines (REH and 697) were treated with RPMI, SCM, or ACM for 24 h and the protein levels of AKT and ERK (phospho- and total) were determined via western blot analysis. Means + s.d. are shown for **a**, **b**, **c** and **d**. *$P < 0.05$, **$p < 0.01$, ***$p < 0.001$, and ****$p < 0.0001$, one-way ANOVA with Tukey's post-test, $n = 3$ independent experiments for **a**, **b**, **c**, and **d**. A representative of $n = 3$ experiments is shown in **e**. Western blot source data are provided in the Source Data file.

adipose-rich microenvironment prior to and during chemotherapy administration. To mimic this scenario in vitro, human B-ALL cell lines were pre-treated for 24 h with RPMI (unconditioned medium), SCM, or ACM prior to the addition of chemotherapies commonly used in the treatment of B-ALL. As expected, methotrexate (MTX), doxorubicin (DOX), and vincristine (VIN) each induced significant cytotoxicity in human B-ALL cell lines in unconditioned medium, and drug-mediated cytotoxicity was not impacted by pre-conditioning with SCM (Fig. 3a–d). Strikingly, chemotherapy-induced cell death for each drug tested was

significantly decreased by 20–50% in cultures pre-treated with ACM (Fig. 3a–d). Similar to chemoresistance induced in human B-ALL cells by in vitro differentiated adipocytes, conditioned medium (CM) from primary adipocytes also significantly reduced MTX-mediated cytotoxicity by 20–40% (Supplementary Fig. 4a, b). In contrast, significant leukemia cell death was observed in human B-ALL cells cultured in fibroblast and osteoblast CM after MTX treatment (Supplementary Fig. 4c, d). In all, these data corroborate previously published studies demonstrating adipocyte-induced chemoresistance to multiple anti-leukemia agents, including

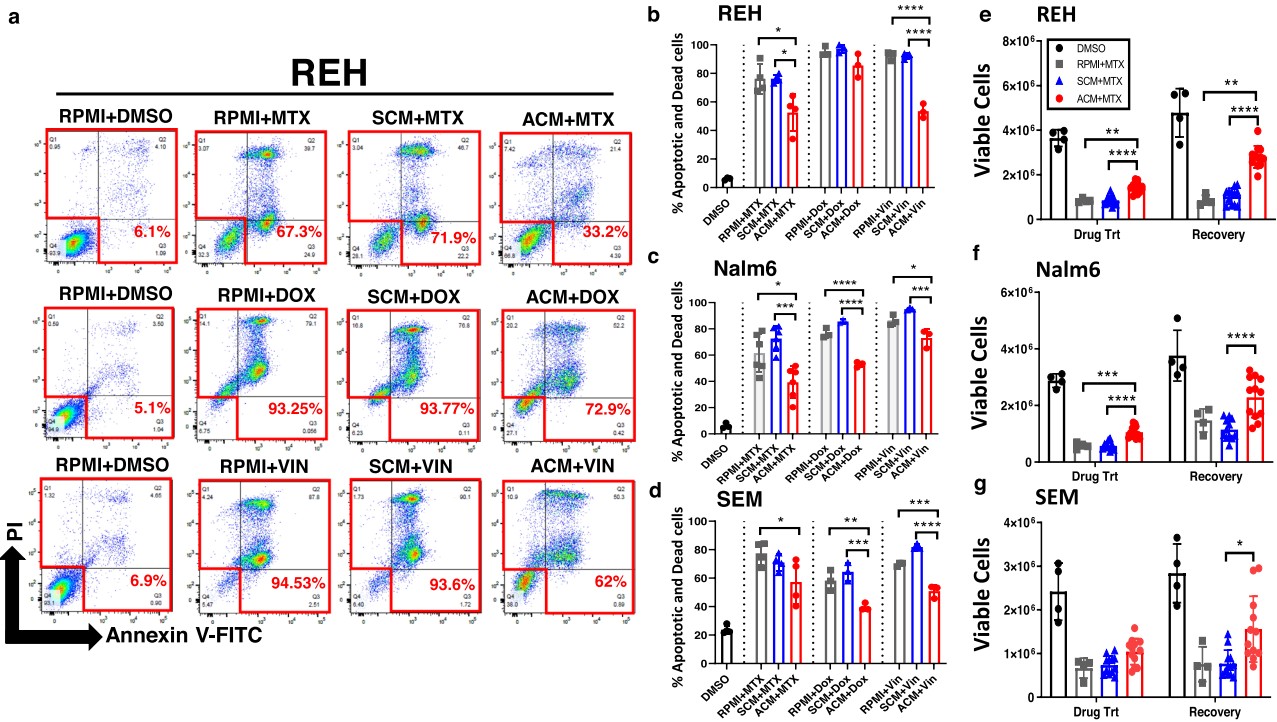

**Fig. 3 Adipocyte-secreted proteins induce chemoresistance in human B-ALL cells.** Human B-ALL cell lines (REH, Nalm6, and SEM) were preconditioned for 24 h with unconditioned medium (10% RPMI), SCM, or ACM prior to treatment with vehicle (DMSO), methotrexate (MTX), doxorubicin (DOX) or vincristine (VIN) for an additional 2 days (cells remained in CM during chemotherapy treatment). **a** Representative primary data for REH cells are shown. **b** The percentage of apoptotic and dead cells on day 3 of culture for each cell line was determined using Annexin-V/ PI staining followed by flow cytometric analysis. **e–g** B-ALL cells were cultured in RPMI, SCM, or ACM and then treated with MTX as described in **a–d** (Drug Trt). On day 3 of culture, cells were harvested, and equal numbers of viable cells were cultured in MTX-free medium for an additional 3 days (Recovery). Viable cells were identified and enumerated using trypan-blue exclusion assays. Means ± s.d. are shown (**$p < 0.01$, ***$p < 0.001$, and ****$p < 0.0001$, $n = 3$ independent experiments, one-way ANOVA with Tukey's post-test).

nilotinib[15], dexamethasone[15], vincristine[15,51], daunorubicin[51,52], and L-asparaginase[14] in B-ALL cells.

To determine the nature of the adipocyte-secreted factor that induced chemoresistance, human B-ALL cells were conditioned with nuclease, lipase, or protease-treated RPMI, SCM, or ACM prior to exposure to MTX. Protease treatment of ACM abolished its ability to confer chemoresistance to human B-ALL cells (Supplementary Fig. 5a-c). In contrast, treatment with nucleases or lipases had no significant impact on chemo-protection conferred by ACM (Supplementary Fig. 5d, e). To determine if direct contact with adipocytes was capable of conferring chemoresistance to B-ALL cells, human B-ALL cells were co-cultured with BMSCs or adipocytes (ADP) for 3 days in the presence of MTX. Similarly to ACM, co-culturing B-ALL cells with adipocytes, but not BMSC, conferred chemoresistance to MTX (Supplementary Fig. 6). Furthermore, the degree of protection was comparable to that observed in human B-ALL cells cultured in ACM (Fig. 3a–d). These results suggest that adipocyte-secreted proteins are largely responsible for inducing chemoresistance in human B-ALL cells. To determine if the surviving B-ALL cells maintain proliferative capacity when the drug was no longer present, human B-ALL cells were pre-conditioned with RPMI, SCM, or ACM and treated with MTX and then equal numbers of viable cells were cultured in RPMI without MTX for an additional 3 days. In these studies, cell density was significantly increased after removal of MTX in cultures of ACM-conditioned leukemia cells relative to cells conditioned with RPMI or SCM (Fig. 3e–g). Thus, ACM-exposed B-ALL cells are more resistant to the cytotoxic effects of chemotherapies commonly used for treatment of B-ALL and have an enhanced ability to proliferate when chemotherapy treatment is discontinued.

**ACM induces global gene expression changes in B-ALL cells.** To address the mechanism of ACM-induced chemoresistance, we explored gene expression changes in human B-ALL cells exposed to RPMI, SCM, and ACM in the presence and absence of MTX. Principal component analysis (PCA) of RNA-sequencing results revealed that culturing human B-ALL cells in the various conditioned media elicited distinct gene expression profiles, which were further altered when cells were treated with the chemotherapy methotrexate (Fig. 4a–d). In addition to PCA analysis, unsupervised hierarchical clustering revealed further insight into conditioned medium-mediated modulation of gene expression changes in human B-ALL cells. As demonstrated by PCA analysis, culturing human B-ALL cells in ACM induced a unique gene expression profile relative to signatures observed when cells were cultured in unconditioned medium or SCM, with these groups exhibiting similar genetic profiles (Fig. 4e). Furthermore, without drug exposure, gene profiles were distinct between RCH-AcV and REH cells cultured in ACM (Fig. 4e). Remarkably, these responses changed dramatically when B-ALL cells were treated with methotrexate in the presence of adipocyte secreted factors, where a subset of genes increased and another subset decreased in both human B-ALL cell lines tested (Fig. 4f and Tables 1–2).

**Adipocytes upregulate Galectin-9 on B-ALL cells.** Among candidates identified through RNA sequencing, we assessed the expression of 20 candidate molecules on 8 human B-ALL cell lines (Table 2) in ACM, including GAL-9, due to its known role in promoting cellular aggregation and a recently described role in acute myeloid leukemia[53]. Only GAL-9 exhibited increased gene and surface expression in every human B-ALL cell line tested, and

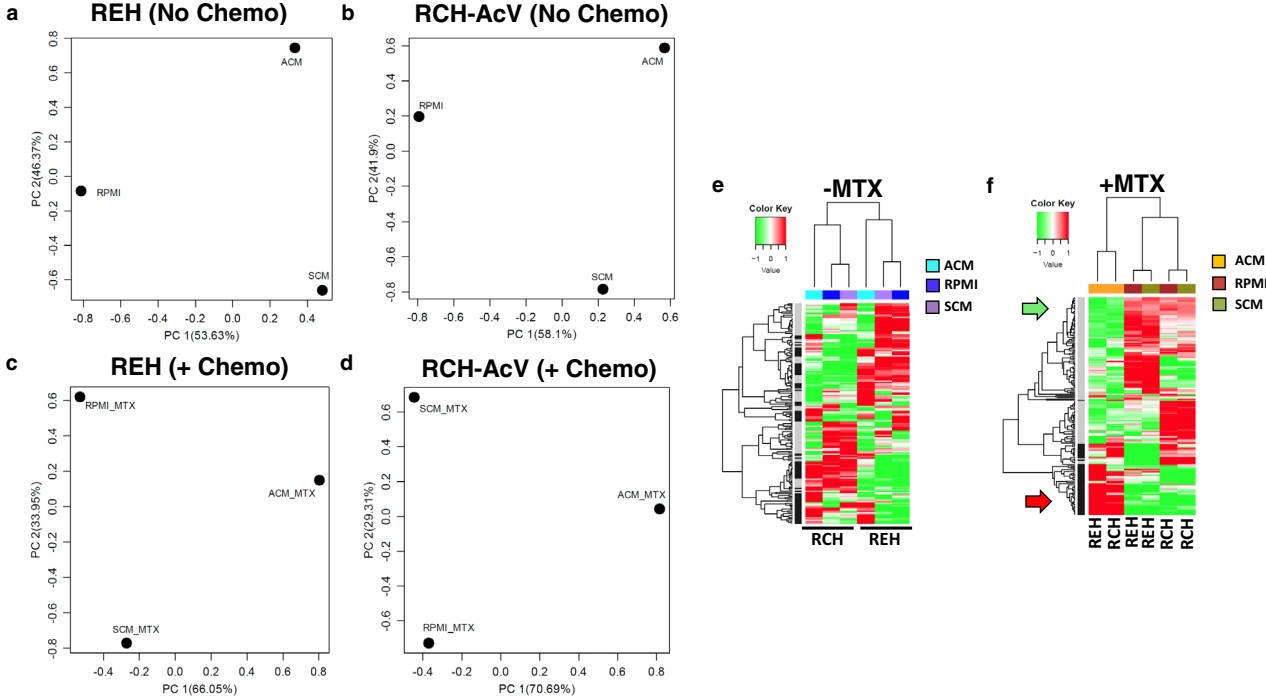

**Fig. 4 Adipocyte-secreted factors induce significant gene expression changes in human B-ALL cells in the absence and presence of methotrexate.**
Human B-ALL cell lines (REH and RCH-AcV) were cultured with ACM for 48 h or preconditioned with ACM for 24 h and cultured with MTX for an additional 24 h (48 total hours of culture). After 48 h of culture, B-ALL cells were harvested, and RNA isolated for RNA-sequencing analysis. **a–d**, RNA-sequencing results were analyzed using principal component analysis (PCA). **e–f**, Unsupervised hierarchical clustering analysis and heatmaps of the RNA-sequencing results are shown with genes upregulated in ACM indicated in black (along with red arrow) and genes downregulated in ACM indicated in gray (along with the green arrow). Treatment conditions are color-coded at the top of the heatmap.

this was induced by both adipocyte-secreted factors (Fig. 5a, b and Supplementary Fig. 7a, b) and in adipocyte co-culture assays (Fig. 5c and Supplementary Fig. 7c). Given that ACM alone was sufficient to induce GAL-9 surface expression on human B-ALL cells, we next sought to identify specific adipocyte-secreted cytokines, which modulated the expression of this lectin. Based on our Luminex and ELISA analyses of the adipocyte and stromal cell secretomes (Supplementary Fig. 2a–c), we chose to determine whether GAL-9 surface expression on human B-ALL cells was regulated by cytokines secreted at significantly higher levels by adipocytes than by bone marrow stromal cells (Supplementary Fig. 2a-c and 8a). To this end, we treated several human B-ALL cell lines with increasing doses of TNF-α, IL-1β, IL-6, and IP-10. Of these, only TNF-α was found to upregulate GAL-9 surface expression on human B-ALL cells (Fig. 5d, Supplementary Fig. 8b, c). This observation is in line with TNF-α mediated upregulation of Galectin-9 in astrocytes and synovial fibroblasts[54,55]. Despite administration of recombinant TNF-α being sufficient to induce GAL-9 surface expression on human B-ALL cell lines, neutralization of TNF-α from ACM resulted in only a modest decrease in GAL-9 surface expression on B-ALL cells which supports the likelihood that multiple adipocyte-secreted factors contribute to the upregulation of GAL-9 surface expression on B-ALL cells (Supplementary Fig. 8d–f). Overall, these results demonstrate that adipocyte-secreted factors, including TNF-α, induce GAL-9 surface expression on human B-ALL cells.

To determine whether the increased surface expression of GAL-9 required continued exposure to adipocyte-secreted factors, human B-ALL cells were cultured in unconditioned medium or CM for 3 days, washed, and cultured in unconditioned for an additional 3 days with GAL-9 surface expression assessed before and after the removal of CM. As previously

observed, culturing human B-ALL cells in ACM significantly induced GAL-9 surface expression relative to levels observed on leukemia cells cultured in unconditioned medium or SCM (Fig. 5e). However, removing the conditioned medium and re-plating the cells in unconditioned medium resulted in GAL-9 surface levels returning to baseline (surface levels observed on human B-ALL cells cultured in RPMI) in each of the cell lines tested (Fig. 5f). Exposing human B-ALL cells to ACM did not alter the baseline surface expression of T-cell immunoglobulin and mucin-domain containing-3 (TIM-3), the canonical receptor for GAL-9 (Supplementary Fig. 9a); however, ACM induced significant colocalization of GAL-9 and TIM-3 on human B-ALL cells (Supplementary Fig. 9b).

Given the strong induction of GAL-9 in ACM-exposed human B-ALL cell lines, we next mined publicly available databases to examine the expression levels of *LGALS9* (the gene encoding GAL-9) and *HAVCR2* (the gene encoding TIM-3) in leukemia cells from patients at diagnosis and relapse. At diagnosis, the expression of *LGALS9* is increased in acute myeloid leukemia (AML), B-ALL, and mixed lineage leukemia (MLL) compared to gene expression levels found in normal peripheral blood mononuclear cells (PBMCs), with samples from patients with B-ALL expressing (on average) the highest *LGALS9* levels (Supplementary Fig. 7d). Further analyses of B-ALL subtypes revealed varying but consistently elevated levels of *LGALS9* expression in samples from patients with different subtypes of B-ALL relative to levels found in normal PBMCs (Fig. 5g). In contrast, *HAVCR2* was expressed at lower levels in leukemia patient samples than observed in PBMCs (Supplementary Fig. 9c). Additionally, high levels of *LGALS9* expression were associated with significantly decreased overall survival in patients with relapsed B-ALL (Fig. 5h).

To examine the impact of body mass index (BMI) on GAL-9 and TIM-3 expression, RNA and surface protein levels were

**Table 1 Genes associated with chemoresistance in both human B-ALL cell lines.**

Genes associated with chemoresistance in both human B-ALL cell lines. REH & RCH-AcV (ACM + MTX); p-value cutoff of 0.01

| Upregulated | Downregulated |
|---|---|
| RSPRY1 | CCDC17 |
| RAD17 | LINC00176 |
| SDF2L1 | IK |
| VPS9D1-AS1 | NSMCE2 |
| OXNAD1 | MYO5C |
| STAT6 | ASTN2 |
| MIR7161 | RAN |
| MCMBP | MORF4L2-AS1 |
| LINC00624 | THOP1 |
| ODF2L | CDS2 |
| TSC1 | MOGS |
| LINC00562 | AFF4 |
| PPARGC1B | 'PIK2IP1 |
| CD3EAP | NAIP |
| SLCO4A1-AS1 | ZFP36L1 |
| ARRDC4 | RAB15 |
| RCN1 | TMEM217 |
| LINC013355 | PEX19 |
| TMEM88 | LIG1 |
| MYD88 | PHLDA3 |
| CNOT8 | GADD45A |
| CHRNA5 | RHEB |
| FMNL2 | TSGA10 |
| HLA-E | |
| CRACR2A | |
| CDNF | |
| LINC01144 | |
| RHNO1 | |
| XYLT1 | |
| EIF5AL1 | |
| PDIA5 | |
| DDN | |
| EFNA3 | |
| SLC25A16 | |
| TLCD1 | |
| GPR137C | |
| ZFP36L2 | |

This table contains genes identified as being associated with chemoresistance in both cell lines using a p-value cutoff of 0.01. Notably, this list did not contain many mediators of cellular aggregation or adhesion, which was a prominent phenotype observed in human B-ALL cells cultured in adipocyte-conditioned media.

**Table 2 Genes associated with chemoresistance in both human B-ALL cell lines.**

Genes associated with chemoresistance, signaling, and adhesion in both human B-ALL cell lines. REH & RCH-AcV (ACM + MTX); p-value cutoff of 0.05

| Upregulated |
|---|
| LGALS9 |
| ST3GAL6 |
| LGALSL |
| LGALS1 |
| LGALS3BP |
| LGALS8 |
| CD44 |
| CD69 |
| ICAM1 |
| ICAM2 |
| ICAM3 |
| ICAM4 |
| ICAM5 |
| PCDH1 |
| PCDHGC3 |
| PCDH9 |
| CDH24 |
| CDH2 |
| ITGA4 |
| ITGB1BP1 |

This table contains genes identified as being associated with chemoresistance in both cell lines using a p-value cutoff of 0.05. The p-value cutoff of 0.05 was used to nominate upregulated genes associated with adhesion and aggregation which could be involved in the induction of chemoresistance. We chose to focus on Galectin-9 in this study because the LGALS9 gene validated in several human B-ALL cell lines and its surface expression was the only one consistently upregulated on human B-ALL cells (n = 8) cultured in adipocyte-conditioned medium.

determined on PBMCs collected from lean patients and patients with obesity at diagnosis of leukemia (Table 3). Samples from patients with obesity had significantly higher levels of the short isoform of the LGALS9 mRNA compared to samples from lean patients with B-ALL (Supplementary Fig. 7g). In contrast, the expression of HAVCR2 was not significantly different in B-ALL cells from lean patients and patients with obesity (Supplementary Fig. 9d). In addition, both the percentage of B-ALL cells with surface expression of GAL-9 and the average level of GAL-9 surface expression were significantly increased on B-ALL cells from patients with obesity relative to B-ALL cells from lean patients (Fig. 5i, j, Supplementary Fig. 7h). In contrast, TIM-3 surface expression was not significantly different on primary B-ALL cells isolated from lean patients and patients with obesity (Supplementary Fig. 9e), consistent with observations in B-ALL cell lines cultured with ACM relative to unconditioned medium and SCM (Supplementary Fig. 9a). To determine if the GAL-9 surface levels differed on B-cells isolated from individuals without leukemia and those with B-ALL, we performed linear regression analysis to determine if there was a relationship between GAL-9 surface expression on B-cells and BMI (Fig. 5k and Table 4). In these studies, we found a significant correlation between GAL-9 surface expression and BMI on smaller, less granular CD19-expressing B-cells (Fig. 5k, Supplementary Fig. 7e, f); however, this relationship was less apparent in larger, more granular CD19-expressing B-cells (Supplementary Fig. 7e, f, Supplementary Fig. 7i). In all, these data demonstrate that adipocytes directly increase GAL-9 expression on human B-ALL cells, which is also elevated on the surface of leukemia cells isolated from patients with obesity.

**αGAL-9 antibodies are cytotoxic to B-ALL**. The significant upregulation of GAL-9 on the surface of human B-ALL cells are coincident with the induction of chemoresistance in the presence of adipocytes or ACM. Furthermore, GAL-9 is significantly increased on the surface of B-ALL cells from patients with obesity and poorer prognoses are observed in patients with B-ALL expressing high levels of GAL-9, suggesting that GAL-9 functions to promote B-ALL survival in the context of obesity. To test this hypothesis, we determined whether treatment with an anti-Galectin-9 (αGAL-9) antibody altered B-ALL phenotypes and viability.

Galectin-9 is a lectin that promotes cellular adhesion[56]. In non-malignant B-cells, GAL-9 also organizes the immunoglobulin M (IgM) and B-cell Receptor (BCR) into large clusters and facilitates their ligation with the inhibitory molecules CD45 and CD22, thus attenuating B-cell activation[57,58]. We found that αGAL-9 antibody treatment prevented ACM-induced cellular aggregation in every human B-ALL cell line tested with the most notable inhibition of aggregation occurring in REH cells (Fig. 6a). Furthermore, treatment with αGAL-9 antibody in the presence of

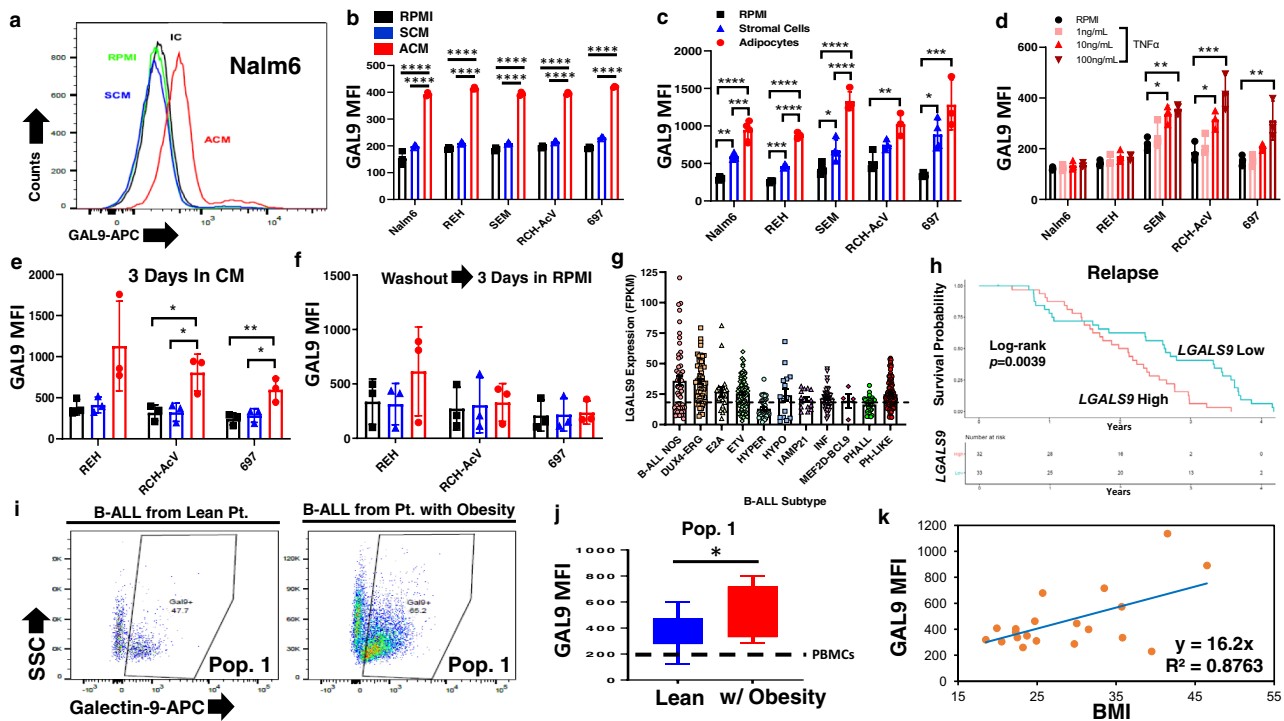

**Fig. 5 Adipocytes induce Galectin-9 surface expression on human B-ALL cells. a**, **b** Human B-ALL cell lines were cultured in unconditioned medium (RPMI), SCM, or ACM for 24 h. **c** In parallel experiments, leukemia cells were co-cultured with stromal cells (SC) or adipocytes (ADP) for 3 days. **a–c** GAL-9 surface expression was determined via flow cytometry. **d** Human B-ALL cell lines were cultured with control Ig in RPMI or recombinant TNF-α for 3 days. The surface expression of GAL-9 was determined as described in **a–c**. **e,f** Human B-ALL cell lines were cultured in RPMI, SCM, or ACM for 3 days. **e** A portion of cells were stained for GAL-9 surface expression. **f**, An equal number of viable cells ($5 \times 10^5$ cells/well) were re-plated for 3 days in conditioned medium before determining GAL-9 surface expression using flow cytometric analyses. **g** *LGALS9* gene expression levels in patient samples with the indicated B-ALL subtypes were determined using the St. Jude PeCan Data Portal. The dotted lines indicate expression levels in PBMCs from lean donors obtained from the EMBL-EBI Expression Atlas (Genotype-Tissue Expression [GTEx] Project; $n = 454$). **h** Patient survival with relapsed B-ALL expressing low or high levels of *LGALS9* was determined using the TARGET dataset. "Low" denotes patients with *LGALS9* levels below the median and "high" denotes above the median. **i-j**, PBMCs were obtained from lean pediatric patients ($n = 8$) and pediatric patients with obesity (w/ obesity; $n = 8$) with B-ALL. **i,j** Flow cytometric primary data are shown for GAL-9 surface expression on leukemic blasts expressing low levels of CD19 (Pop. 1) (Supplementary Fig. 6 has gating strategies and Table 1/2 contain patient demographics). The dotted line in j represents the average GAL-9 MFI on non-malignant B-cells from healthy donors. Boxplots display the 25th and 75th percentile. The whiskers extend from the minima to the maxima, respectively. **k**, The relationship between BMI and surface GAL-9 expression on CD19$^+$ cells (normal cells and leukemia blasts) was determined for Pop. 1. Means ± s.d. are shown in **b–g** and **j**. Statistical significance is denoted by *$p < 0.05$, **$p < 0.01$, ***$p < 0.001$, ****$p < 0.0001$. $n = 5$ independent experiments, one-way ANOVA with Tukey's post-test (**b-f**) or two-sided Student's *t*-test (**j**)).

ACM induced a significant accumulation of B-ALL cells in G2/M phases of the cell cycle (Fig. 6b–d). Altered cell cycle progression was accompanied by a complete ablation (Cyclin D3 and CDK4) or reduced protein expression (Cyclin A and CDK2) of cell cycle regulators, whereas E2F1 protein levels (which promote cell cycle progression) were increased (Fig. 6e, f and Supplementary Fig. 10a–d). The accelerated cell cycle progression of αGAL-9 antibody-treated, adipocyte-exposed human B-ALL cells was accompanied by increased DNA damage (elevated γH2AX and cPARP levels), which was significantly less extensive when human B-ALL cells were treated with αGAL-9 antibody in the presence of unconditioned medium or SCM (Fig. 6e, f, Supplementary Fig. 11a). These results demonstrate that GAL-9 is a critical cell cycle regulator in B-ALL cells after ACM exposure. Consequently, antibody-mediated inhibition of GAL-9 on ACM-exposed B-ALL cells induces inappropriate cell cycle progression and extensive DNA damage.

We also observed increases in the sub-G1 population in ACM-exposed cells treated with αGAL-9 antibody (Fig. 6b–d), so we next determined if αGAL-9 antibody treatment induced apoptosis in ACM-exposed human B-ALL cells. Caspase 3 activation was

significantly induced in αGAL-9 antibody-treated human B-ALL cells cultured in ACM but not the other conditions tested (Fig. 6g, h). This response preceded the induction of extensive cell death in ACM-exposed B-ALL cells, whereas αGAL-9 antibody treatment was not cytotoxic to human B-ALL cells cultured in unconditioned medium or SCM (Fig. 6i–l, Supplementary Fig. 10e, Supplementary Fig. 11b). To better understand the mode of action of αGAL-9 antibody treatment, we assessed *AKT* and *ERK1* gene expression levels in B-ALL cells cultured in unconditioned medium or ACM. In multiple B-ALL cell lines, we found that αGAL-9 antibody treatment increased *AKT* and *ERK1* gene expression levels more dramatically in leukemia cells cultured in ACM relative to unconditioned medium (Supplementary Fig. 11c–e). The licensing of these pathways suggests that blocking GAL-9 on ACM-exposed B-ALL cells activates these cells, consistent with its recently identified inhibitory role on normal B-cells[57,58]. The increased activation of these pathways in αGAL-9 antibody-treated B-ALL is consistent with our observations of increased cell cycle progression (with accompanying DNA damage, Caspase 3 activation, and cell death). The expression of Galectin-9 in human B-ALL is required for αGAL-9 antibody mediated cell death, where we observed that

**Table 3 Patient and donor demographics for Fig. 5j.**

| Characteristic | All Patients | | BMI-5th and 85th Percentile (calculated using weight/height/age) | | BMI ≥ 95th Percentile (calculated using weight/height/age) | | P value |
|---|---|---|---|---|---|---|---|
| | No. | % | No. | % | No. | % | |
| Gender | | | | | | | >0.999 |
| Male | 9 | 56.0 | 5 | 63.0 | 4 | 50.0 | |
| Female | 7 | 44.0 | 3 | 37.0 | 4 | 50.0 | |
| Age (years) | | | | | | | 0.004 |
| Mean ± SD | 14.2 ± 2.5 | | 12.6 ± 1.6 | 15.9 ± 2.2 | | | |
| Range | 10.3–19.3 | | 10.3–14.6 | 13.2–19.3 | | | |
| Ethnicity | | | | | | | >0.999 |
| Hispanic | 3 | 19.0 | 1 | 12.0 | 2 | 25.0 | |
| Non-Hispanic | 13 | 81.0 | 7 | 88.0 | 6 | 75.0 | |
| Race | | | | | | | 0.549 |
| Caucasian | 9 | 56.0 | 5 | 63.0 | 5 | 63.0 | |
| African American | 5 | 31.0 | 3 | 37.0 | 2 | 25.0 | |
| Other | 1 | 6.0 | 0 | 0.0 | 1 | 12.0 | |
| Initial WBC | | | | | | | 0.569 |
| <50,000 | 12 | 75.0 | 7 | 88.0 | 5 | 63.0 | |
| ≥50,000 | 4 | 25.0 | 1 | 12.0 | 3 | 37.0 | |
| CNS Disease | | | | | | | 0.106 |
| CNS 1 | 11 | 69.0 | 7 | 88.0 | 4 | 50.0 | |
| CNS 2 | 5 | 31.0 | 1 | 12.0 | 4 | 50.0 | |
| CNS 3 | 0 | 0.0 | 0 | 0.0 | 0 | 0.0 | |
| MRD | | | | | | | 0.158 |
| <0.01 | 13 | 81.0 | 8 | 100.0 | 5 | 63.0 | |
| ≥0.01 | 2 | 13.0 | 0 | 0.0 | 2 | 25.0 | |
| N/A | 1 | 6.0 | 0 | 0.0 | 1 | 12.0 | |

This table contains donor demographics with the p-value cutoff of 0.05 which was used to determine significant changes between donor parameters. The body mass index (BMI) was calculated using the Centers for Disease Control and Prevention (CDC) growth charts for children and teens 2–19 years of age (https://www.cdc.gov/healthyweight/bmi/calculator.html).

**Table 4 Patient and donor demographics for Fig. 5k.**

| Characteristic | All Patients | | BMI-5th and 85th Percentile (calculated using weight/height/age) | | BMI ≥ 95th Percentile (calculated using weight/height/age) | | P value |
|---|---|---|---|---|---|---|---|
| | No. | % | No. | % | No. | % | |
| Leukemia Present | | | | | | | 0.361 |
| Yes | 10 | 50.0 | 3 | 37.0 | 7 | 58.0 | |
| No | 10 | 50.0 | 5 | 63.0 | 5 | 42.0 | |
| Age (years) | | | | | | | 0.575 |
| Mean ± SD | 19.4 ± 6.9 | | 18.3 ± 5.9 | | 20.1 ± 7.7 | | |
| Range | 11.5–36 | | 11.5–25 | | 13.2–36 | | |
| Gender | | | | | | | 0.848 |
| Male | 7 | 35.0 | 3 | 38% | 4 | 33% | |
| Female | 13 | 65.0 | 5 | 62% | 8 | 67% | |
| Race | | | | | | | 0.201 |
| Caucasian | 13 | 65.0 | 7 | 88.0 | 6 | 50.0 | |
| African American | 5 | 25.0 | 1 | 12.0 | 4 | 33.0 | |
| Other | 2 | 10.0 | 0 | 0.0 | 2 | 17.0 | |

This table contains donor demographics with the p-value cutoff of 0.05 which was used to determine significant changes between donor parameters. The body mass index (BMI) was calculated using the Centers for Disease Control and Prevention (CDC) growth charts for children and teens 2–19 years of age (https://www.cdc.gov/healthyweight/bmi/calculator.html).

knocking down GAL-9 in human B-ALL cells prevented antibody-induced apoptosis (Supplementary Fig. 11f, g). Interestingly, knocking down GAL-9 in human B-ALL cells induces more cell death when leukemia cells were cultured in ACM relative to unconditioned medium (Supplementary Fig. 11g). Reduced GAL-9 levels in human B-ALL cells results in a failure to undergo senescence when cultured in ACM, which demonstrates that GAL-9 is required for this response (Supplementary Fig. 11h). Although inhibiting GAL-9 abrogates senescence and is cytotoxic to ACM-exposed human B-ALL cells, treating human B-ALL cells with recombinant GAL-9 failed to induce apoptosis when added alone (Supplementary Fig. 11i) or when combined with MTX treatment (Supplementary Fig. 11j). In all, these results highlight a previously unrecognized role for Galectin-9 in promoting the survival of B-ALL cells in adipose-rich microenvironments and demonstrates that antibody-mediated blockade of this lectin in ACM-exposed human B-ALL cells results in significant apoptosis in vitro.

**αGAL-9 immunotherapy protects obese mice from B-ALL.** Given the extensive cytotoxicity observed in ACM-exposed B-ALL cells treated with αGAL-9 antibody, we postulated that combining chemotherapy and αGAL-9 antibody treatment, which demonstrated efficacy in murine models of *Mycobacterium tuberculosis*[59], would extend the survival of obese mice with B-ALL. Prior to conducting in vivo experiments, human B-ALL cells were cultured in unconditioned medium, SCM, or ACM and treated with MTX, αGAL-9 antibody, or their combination to assess apoptosis in vitro. Similar to previous observations, culturing human B-ALL cells in ACM, but not SCM, resulted in significant chemoresistance to MTX (Supplementary Fig. 12a). However, combining αGAL-9 antibody and MTX treatment resulted in greater than 90% cell death in human B-ALL cells cultured in ACM (Supplementary Data Fig. 12a).

A murine BCR-ABL+ *Arf*−/− B-ALL (mB-ALL) model was used to test whether treatment with αGAL-9 antibody could be used as a therapy to treat obese mice with B-ALL. Similar to human B-ALL cells, we found that ACM also induced

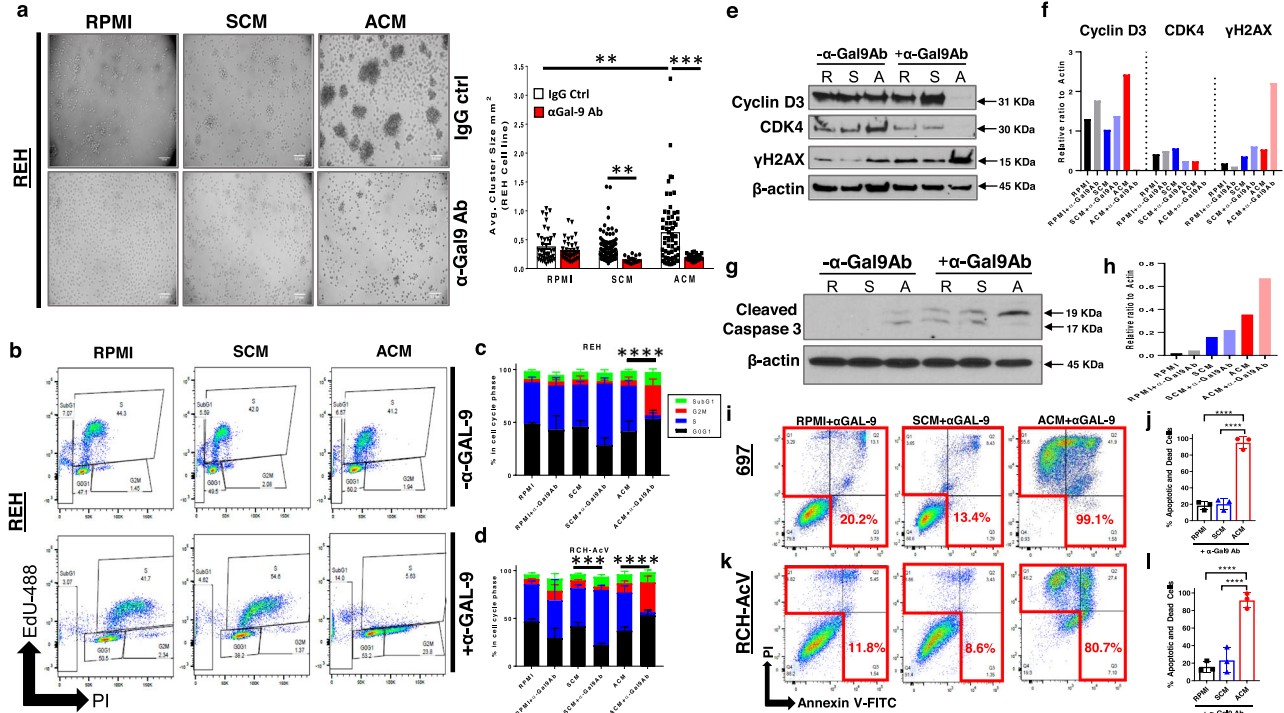

**Fig. 6 Treatment with αGalectin-9 antibody alters cell cycle progression and induces cell death in human B-ALL cells in the presence of adipocyte-secreted factors. a** Human B-ALL cell line REH was cultured for 24 h in unconditioned medium (RPMI), SCM, or ACM and then treated with αGalectin-9 (αGAL-9) or IgG control (IgG ctrl) antibody for an additional 2 days. On day 3 of culture, brightfield images were obtained and cell cluster sizes were quantified. **b-d**, REH cells were treated with IgG (no αGAL-9 antibody) or αGAL-9 antibody for 24 h. After a day of treatment, 5-ethynyl-2'-deoxyuridine (EdU) was added to cultures for 2 h and then cells were permeabilized and stained with propidium iodide (PI). The fraction of cells in S (EdU+), G1 (2 N DNA content), and G2/M (4 N DNA content) phases of the cell cycle was determined by flow cytometric analysis. Representative data are shown in b. **e-h**, Lysates were prepared after 48 h of culture in RPMI (R), SCM (S), or ACM (A) with IgG (no αGAL-9 antibody) or αGAL-9 antibody. The presence of Cyclin D3, CDK4, γH2AX, and β-actin (loading control) protein was determined by western blot analysis. Primary data is shown in **e** and **g** and quantitative data is shown in f and h. **i-l**, Human B-ALL cell lines (697 and RCH-AcV) were cultured for 24 h in RPMI, SCM, or ACM and then treated with anti-GAL-9 antibody (αGAL-9 Ab) or an IgG control antibody (IgG ctrl) for an additional 2 days. The percentage of apoptotic and dead cells on day 3 of culture was determined using Annexin-V/ PI staining followed by flow cytometric analysis. Representative data are shown in **i,k**. Means ± s.d. are shown (*p < 0.05, **p < 0.01, ***p < 0.001, and ****p < 0.0001, n = 4 independent experiments, one-way ANOVA with Tukey's post-test). In **b-d** ****p < 0.0001, n = 3 independent experiments, two-sided Student's t-test of the G2/M cell cycle profiles. Western blot source data are provided in the Source Data file.

chemoresistance to methotrexate and doxorubicin in mB-ALL cell cultures (Supplementary Fig. 12b) and upregulated surface Galectin-9 expression when cultured in ACM (Supplementary Data Fig. 12c, d). Furthermore, the percentage of Galectin-9-positive mB-ALL cells significantly increased after their transplantation into obese, but not lean, mice (Supplementary Fig. 12e), which further validated their utility for these studies.

C57BL/6 mice were maintained on lean and high-fat diets as previously described and then challenged with mB-ALL (Fig. 7a). Treatment with vehicle, MTX, αGAL-9 antibody, or a combination of MTX and αGAL-9 antibody was initiated on day 7 after leukemia inoculation, when GFP+ B-ALL cells were detected in the peripheral blood. All drug combinations were well-tolerated in lean mice; however, obese mice loss substantial weight when treated with MTX alone (Supplementary Fig. 12f). The significant amount of weight loss observed in obese mice was mitigated with single-agent αGAL-9 antibody treatment alone and when combined with MTX administration (Supplementary Fig. 12f), suggesting that αGAL-9 antibody treatment strategies improve drug tolerability in obese mice with B-ALL.

We found that MTX significantly extended the survival of lean mice challenged with mB-ALL relative to mice receiving the vehicle control (Fig. 7b). In contrast, αGAL-9 single-agent treatment did not significantly extend survival in lean mice relative to vehicle-treated mice (Fig. 7b). Furthermore, the survival of lean mice receiving the combination treatment of MTX and αGAL-9 antibody was equivalent to lean mice receiving the single-agent MTX regimen (Fig. 7b).

Similar to the outcomes shown in Fig. 1c, leukemogenesis was accelerated in mice fed high-fat diets, as indicated by the observation that vehicle-treated obese mice succumbed to mB-ALL by day 17 post-challenge, whereas vehicle-treated lean mice succumbed to mB-ALL by day 29 post-challenge (Fig. 7b, c). In contrast to lean mice, treatment with single-agent αGAL-9 antibody significantly prolonged survival in obese mice with approximately 60% of mice surviving past 3 months of challenge with mB-ALL (Fig. 7b, c). Unlike the protection observed in lean mice treated with MTX, obese mice treated with MTX were not significantly protected relative to vehicle-treated mice (Fig. 7b, c). The combination of MTX and αGAL-9 antibody treatment in obese mice with mB-ALL did not augment survival over single-agent αGAL-9 antibody treatment, suggesting that the significant extension in survival observed in obese mice was mainly attributable to αGAL-9 antibody treatment (Fig. 7c). The survival of lean and obese immunocompromised mice transplanted with human B-ALL cells was also determined after treatment with vehicle, MTX, αGAL-9 antibody, or a combination of MTX and αGAL-9 antibody. The outcomes of the xenograft experiments

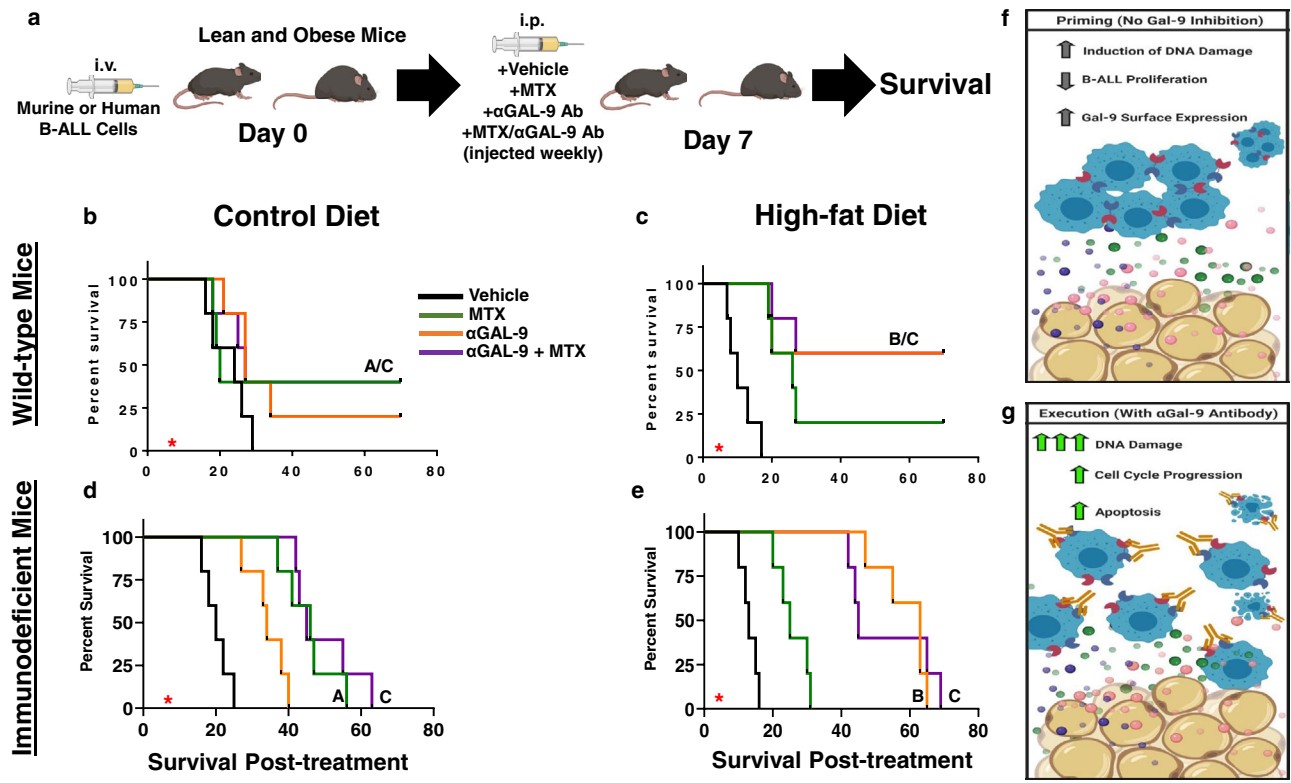

**Fig. 7 αGalectin-9 treatment prolongs survival in obese mice challenged with B-ALL. a** Lean and obese C57BL/6 wild-type or NOG immunocompromised mice (2 months) were inoculated intravenously with $2 \times 10^5$ murine Bcr-Abl[+] Arf−/− B-ALL cells (mB-ALL) or REH cells, respectively. On day 7 postinoculation, when GFP[+] leukemic blasts were detected in the peripheral blood, treatment with DMSO (Vehicle, $n = 5$/group), 0.75 mg/kg methotrexate (MTX, $n = 5$/group), 1.5 mg/kg αgalectin-9 antibody (αGAL-9 Ab, $n = 5$/group), or MTX + αGAL-9 Ab ($n = 5$/group) combined was initiated. Treatments were administered once weekly. **b–e**, Survival was monitored, and significance was determined using the log-rank test with A representing a statistically significant difference between vehicle and MTX-treated mice, B between vehicle and αGal-9 treated groups, and C between vehicle and MTX + αGal-9 treated mice. **f** Proposed model of how adipose-rich microenvironments prime human B-ALL cells for cell death, which is inhibited by the upregulation of Galectin-9 (which acts as a cellular brake). **g** When Galectin-9 is inhibited with αGAL-9 antibody treatment (essentially inhibiting the actions of the brake), cell cycle progression, DNA, and caspase 3 activation are induced resulting in substantial B-ALL cytotoxicity.

corroborated observations made in syngeneic experiments, where MTX treatment was superior in lean mice, αGAL-9 antibody treatment led to significant survival outcomes in obese mice, and the combination of both treatments did not further increase survival in either context (Fig. 7d, e). Unlike experiments conducted in wild-type mice, all immunocompromised mice succumbed to leukemia (Fig. 7d, e), highlighting the importance of the adaptive immune system in the long-term control and eradication of B-ALL cells.

In summary, these observations highlight that adipocyte-induced "priming" of B-ALL cells can be exploited to induce extensive apoptosis when the GAL-9 mediated "brake" is inhibited (Fig. 7f, g). Effectively inhibiting this mechanism in B-ALL cells in adipose-rich microenvironments represents an unexplored therapeutic approach as an alternative treatment strategy for a subset of patients with B-ALL.

## Discussion
In these studies, we show that in the absence of chemotherapy treatment, adipocytes promote the aggregation of human B-ALL cells and induce cellular senescence, which coincides with the activation of senescence-promoting mediators (AKT, BCL-xL, ERK, and XIAP). Overall, our data suggest that human B-ALL cells respond in this manner to the adipocyte secretome to avoid genotoxic-induced cell death. In addition to mitigating the effects of microenvironmental-induced cell death, these responses also

render human B-ALL cells more resistant to chemotherapy-induced cell death, given that these drugs are cytotoxic to rapidly dividing cells.

To identify mechanisms of adipocyte-mediated chemoresistance, we performed RNA-sequencing analysis on human B-ALL cells cultured in unconditioned medium, SCM, and ACM in the absence and presence of methotrexate (MTX). By comparing gene expression changes in two distinct human B-ALL cell lines, we were able to identify GAL9 as a candidate molecule contributing to the observed phenotypes. In the absence of drug treatment, distinct gene expression changes were observed in human B-ALL cells cultured under each condition, with unique gene expression profiles apparent in individual cell lines. Interestingly, when leukemia cells were cultured in MTX, overlapping gene expression profiles were observed in B-ALL cells cultured in ACM relative to the other conditions tested. From these observations, we posited that the adipocyte secretome induced the expression of genes critical for evading chemotherapy-mediated cytotoxicity. Furthermore, given our observation of extensive B-ALL clustering in ACM, we hypothesized that genes coding for proteins that promote cellular adhesion and survival were the best candidates to study to identify mechanisms of adipocyte-mediated chemoresistance.

After mining our data for candidate genes that fit both criteria, we found that the induction of chemoresistance by adipocyte-secreted proteins was coincident with the induction of the *LGALS9* gene expression. This gene codes for the lectin,

Galectin-9 (GAL-9), which we also found to be upregulated on the surface of multiple human B-ALL cell lines when cultured in ACM. Furthermore, this relationship was observed in patients with leukemia, where GAL-9 surface expression was significantly increased on B-ALL cells isolated from patients with obesity relative to lean patients. The analysis of GAL-9 surface levels on B-ALL samples isolated from pediatric patients revealed a significant positive correlation with increasing BMI.

These observations resulted in several questions, including 1) what were the adipocyte-secreted factors, which upregulate GAL-9 on the surface of human B-ALL cells, 2) how did GAL-9 impact the function of human B-ALL in adipose-rich microenvironments, 3) did GAL-9 require binding partners to impact the function of B-ALL cells, and 4) would targeting GAL-9 have therapeutic implications for B-ALL pathogenesis?

Using Luminex and ELISA analyses, we identified several cytokines and chemokines which were secreted at significantly higher levels by adipocytes relative to bone marrow stromal cells. These included previously described cytokines associated with adipocyte-mediated chronic inflammation such as IL-1β, IL-6, and TNF-α[3], as well as, the chemokine IP-10 which has been shown to be secreted by adipocytes in a few studies[60–62]. Of these, we found that only TNF-α exposure could induce GAL-9 surface expression on human B-ALL cells when added as a single cytokine. However, when TNF-α was neutralized from ACM, GAL-9 surface expression remained high on adipocyte-exposed B-ALL cells. These results demonstrate that TNF-α signaling is sufficient to induce GAL-9 surface expression on human B-ALL cells; however, it is not required in the context of the complex adipocyte secretome where additional factors are also capable of upregulating the surface expression of this lectin. Further investigation is warranted to identify other adipocyte-secreted factors which induce GAL-9 surface expression on malignant B-cells.

Human B-ALL cells cultured in adipocyte-secreted factors exhibited several phenotypes, which we posit are related to the increased surface expression of GAL-9. GAL-9 signaling in AML cells induces the activation of AKT and ERK signaling pathways[53], which are also induced in human B-ALL cells after exposure to adipocyte-secreted factors. We posit that GAL-9 is upregulated on human B-ALL cells in adipose-rich microenvironments to prevent apoptosis resulting from microenvironment-induced genotoxic stress. In non-malignant B-cells, GAL-9 suppresses B-cell activation by coupling activating receptors (e.g., B-cell receptor and IgM) with inhibitory receptors (e.g., CD45 and CD22) on the cell surface[57,58]. Our results suggest that GAL-9 also suppresses the function of human B-ALL cells by serving as a cell cycle checkpoint regulator, which reduces the proliferation or induces senescence in malignant B-cells attempting to repair damaged DNA (see Fig. 7f). The acquisition of senescence serves as a DNA damage checkpoint in human fibroblasts[63], which demonstrates that this protective response also occurs in non-malignant cells experiencing genotoxic stress.

In support of our hypothesis that GAL-9 acts as a cell cycle regulator in human B-ALL cells, which becomes critical for survival in adipose-rich microenvironments, we found that antibody-mediated inhibition of GAL-9 on ACM-cultured human B-ALL cells resulted in significant changes in cell cycle profiles in B-ALL cells. Specifically, we observed a decrease in the percentage of human B-ALL cells synthesizing DNA and a compensatory accumulation of cells in G2/M, only when leukemia cells were cultured in ACM and not in unconditioned or stromal cell-conditioned medium. Indeed, αGAL-9 antibody treatment of ACM-exposed B-ALL cells suppressed or ablated the protein levels of key cell cycle checkpoint regulators (Cyclin A/Cyclin D3/CDK2/CDK4) except for the cell cycle promoting transcription factor E2F1. Furthermore, increases in cell cycle progression in αGAL-9 antibody-treated B-ALL cells exposed to the adipocyte-secretome were concomitant with augmented cPARP and γH2AX activation (DNA damage), as well as Caspase 3 activation and Annexin V/PI positivity (apoptosis). Increased cell death and a failure to induce senescence were also observed in GAL-9 deficient human B-ALL cells. In all, these data highlight an unappreciated role for GAL-9 as an important cell cycle checkpoint regulator in human B-ALL cells under conditions which induce genotoxic stress (e.g. increased adiposity).

The therapeutic potential of antibody-mediated targeting of GAL-9 was evident in syngeneic and xenograft experiments conducted in lean and obese mice transplanted with murine and human B-ALL cells, respectively. In these experiments, antibody-mediated targeting of GAL-9 significantly extended the survival of obese mice challenged with B-ALL. The extension in survival of obese mice was accompanied by less weight loss compared to obese mice treated with MTX, suggesting both increased efficacy and decreased toxicity associated with αGAL-9 antibody treatment in adipose-rich backgrounds. In contrast, αGAL-9 antibody treatment was not cytotoxic to B-ALL cells cultured in unconditioned medium or SCM and did not significantly extend the survival of lean mice harboring B-ALL cells when administered alone or in combination with MTX.

At this time, it is unclear if this response is dependent on GAL-9/TIM-3 interactions or other ligands reported to be associated with GAL-9, such as CD44, CD40, CLEC7a (Dectin-1), and CD137 (41BB)[64–68]. These questions warrant further investigation to comprehensively determine how GAL-9 regulates the survival of B-ALL cells in adipose-rich microenvironments. Despite a lack of complete understanding of the role of GAL-9 ligands on cell cycle progression or apoptosis, our data demonstrate increased GAL-9/TIM3 localization in ACM-cultured human B-ALL cells. Unlike GAL-9, ACM did not induce significant increases in HAVCR2, the gene encoding TIM3, nor did it increase TIM3 surface expression on human B-ALL cells. These results are consistent with our data mining results from the St. Jude PeCan Data Portal which revealed that the HAVCR2 gene is expressed at low levels in several B-ALL subtypes compared to levels found in healthy PBMCs, as well as our patient data demonstrating similar TIM3 surface levels on human B-ALL cells isolated from lean patients and patients with obesity. Based on these results, we conclude that the major role of TIM3 in human B-ALL cells might be to traffic GAL-9 to the surface of B-ALL cells, which would be consistent with its reported role in AML cells[69].

Despite the potent cytotoxic effects on human B-ALL cells in our in vitro model and extensive protection of mice transplanted with human B-ALL cells, more studies will need to be conducted to fully realize the therapeutic utility of αGAL-9 antibody treatment. For example, our data show that the gene encoding GAL-9, LGASLS9, is highly expressed in human hematopoietic stem cells, NK cells, monocytes, granulocytes, and naïve B-cells (Supplementary Fig. 7j); therefore, future studies should determine if these populations are affected by αGAL-9 antibody treatment. Along these lines, it will be important to determine if the surface expression of GAL-9 on these cells is modulated by the inflammatory microenvironment, during cellular differentiation, or when proliferation is induced which will be useful in defining the safety profiles of antibody-mediated GAL-9 targeting.

Collectively, our results highlight the importance of GAL-9 in leukemia pathogenesis in the context of obesity and demonstrate the utility of a translational agent that targets this surface lectin on B-ALL cells. In addition to our findings, recent reports also demonstrate that GAL-9/TIM-3 interactions support the maintenance of leukemia-initiating cells (LICs), promote immune evasion in AML models, and that GAL-9 is upregulated in AML

patients who fail chemotherapy treatment[53,70–73]. Thus, GAL-9 inhibition represents an unexplored approach to improve outcomes for patients with obesity with B-ALL and other leukemia subtypes.

## Methods

**Cell lines.** Human B-cell acute lymphoblastic leukemia (B-ALL) cell lines were gifted from Dr. Graham and Dr. Porter laboratories (Department of Pediatrics at Emory University School of Medicine). The Nalm6 cell line was grown in RPMI1640 media (cat# 10-040-CV, Corning) supplemented with 10% fetal bovine serum (FBS, cat# S11550, Atlanta Biologicals). REH, SEM, RCH-AcV, and 697 cell lines were grown in RPMI1640 media supplemented with 20% FBS. GFP-expressing murine OP-9 bone marrow stromal cells were maintained in Alpha-Minimum Essential Medium (αMEM, cat# 15-012-CV, Corning) supplemented with 20% FBS. Murine B-ALL cells (mB-ALL cell line; BAML-Mig; GFP-expressing Bcr-Abl+ Arf-null) were grown in RPMI1640 media supplemented with 20% FBS. MC3T3-E1 subclone 4 mouse preosteoblast cell line (ATCC, cat# CRL-2593) was maintained in MEM-alpah (Gibco, cat# A10490-01) media supplemented with 10% FBS and 1X Pen/Strep. 3T3-L1 mouse fibroblast cell line (ATCC, cat# CL-173) was maintained in DMEM supplemented with 10% FBS and 1X Pen/Strep.

**Mice and diet-induced obesity.** Two-month old C57BL/6 mice (The Jackson Laboratory) or NOG mice (Taconic; [genotype] sp/sp/ko/ko [nomenclature] NOD.Cg-Prkdc^scid Il2rg^tm1Sug/JicTac and [genotype] sp/sp;ko/y [nomenclature] NOD.Cg-Prkdc^scid Il2rg^tm1Sug/JicTac) were fed control (10% fat calories) or high-fat (60% fat calories) diets for two to four months prior to experimentation. The obese phenotype was verified in mice prior to experimentation based on the following criteria: significant weight gain, the chronic production of cytokines/chemokines (IL-6; Invitrogen, cat# 88-7064-22 and TNF-α; Invitrogen, cat# 88-7324-22), and elevated insulin levels (RayBioTech, CODE: ELM-Insulin-1). The chow was purchased from Bio-Serv (cat# F4031 for control diets and cat# S3282 for high-fat diets) and sterilized by irradiation prior to usage. Male and female mice were used from this experiment.

**In vivo experiments.** [Syngeneic Studies] Male and female, lean and obese C57BL/6 mice were challenged with mB-ALL i.v. ($10^5$–$10^6$ cells/ mouse). Once signs of leukemia manifested (the detection of GFP+ cells in the peripheral blood, lethargy, ruffled fur, labored breathing, or greater than 7% weight loss), mice (n = 5/group) were then treated with MTX (i.p.; 0.75 mg/kg; weekly), αGal-9 antibodies (i.p.; 1.5 mg/kg; weekly), or the combination therapy. [Xenograft Studies] Lean and obese NOG immunocompromised mice (2 months) were inoculated intravenously with $2 \times 10^5$ REH cells. On day 7 post-inoculation, treatment with DMSO (n = 5), 0.75 mg/kg methotrexate (n = 5), 1.5 mg/kg αgalectin-9 antibody (n = 5), or MTX + αGAL-9 Ab (n = 5) combined was initiated. Treatments were administered once weekly. Survival was monitored pre- and during treatment. The weight of each mouse was recorded prior to treatment and weekly thereafter to determine the extent of weight loss in mice receiving treatment. Bone marrow (BM) was harvested from euthanized moribund mice to determine leukemic burden in the BM at the time of euthanasia (GFP+ B-ALL cells in the BM were detected via flow cytometric analysis). All murine experiments received ethical approval from the Emory University School of Medicine Institutional Animal Care and Use Committee (IACUC) under the approved protocol number DAR-3000013.

**Adipocyte differentiation protocol.** Adipocyte differentiation using murine OP-9 bone marrow stromal cells was performed as previously reported[32–74]. Briefly, confluent cells were trypsinized and plated at $10^5$ cells/well in 6-well plates. Cells were cultured in DMEM media supplemented with 10% FBS. On the following day, media was changed with either fresh DMEM media supplemented with 10% FBS for stromal cell culture or insulin-oleate media (IOM, 1.8 mM Oleate bound to BSA with the molar ratio of 5.5:1) for adipocyte differentiation. Supernatants from bone marrow stromal cells and adipocytes were harvested on days 1, 2, and 3 of culture for use in leukemia experiments.

**Pre-adipocyte isolation from mice.** Pre-adipocytes were isolated from subcutaneous adipose tissue of mice fed high-fat diets. Our protocol was adapted and modified based on previously published procedures (Jun Wu, Thermogenic fat: methods and protocols, pg 3–8, 2017, Methods in Molecular Biology, 1773, 137–146, 2018[75]). Briefly, subcutaneous adipose tissues were dissected from the hind legs. Adipose tissue was washed in 1X PBS, minced into small pieces, and transferred into 50 mL conical tubes containing digestion buffer (5 mL/g tissue, 123 mM NaCl, 5 mM KCl, 1.3 mM CaCl2, 5 mM glucose, 100 mM HEPES, 1X Pen/Strep, 4% BSA (pH 7.45) with 1 mg/mL collagenase II). Tissues were incubated for 1 h at 37 °C with setting at 100 rpm (in an orbital shaker). Digestion was stopped by adding 1/5 vol of FBS, followed by centrifugation at 450 g for 5 min, and the digested solution was filtered using a 100 μm cell strainer. Cell pellets containing preadipocytes (the top "oil" layer contains mature adipocytes) were resuspended in 2 mL ACK red cell lysis buffer (Quality Biological, cat# 118–156-721) by gently vortexing and incubating at room temperature for 3–5 min. ACK lysis buffer was

neutralized by adding 10 mL Hank's balanced salt solution (HBSS) (ThermoFisher Scientific, cat# 14025-076) and cells were collected by spinning at 450 g for 5 min. Pre-adipocytes were counted and plated in 15% FBS DMEM/F-12 media (Bio-Whittaker, cat#12-719 F) and maintained at 37 °C until differentiation. Mature adipocytes were differentiated from pre-adipocytes as described above.

**Co-culturing adipocytes and human B-ALL.** GFP-expressing OP-9 bone marrow stromal cells were used for adipocyte differentiation as previously described. One day post differentiation, $5 \times 10^5$ human B-ALL cells were cultured with $10^5$ bone marrow stromal cells or adipocytes (a 5:1 B-ALL to feeder cell ratio) for 48 h. In cultures treated with MTX, 50 nM of the chemotherapy was added 1 day after the addition of human B-ALL cells. Cell death was determined after 96 h of culture using Annexin-V/PI analysis as described below.

**Oil Red O and Nile Red staining.** To confirm OP-9 cell differentiation into adipocytes, Oil Red O (cat# O1391, Sigma-Aldrich, St. Louis, MO) or Nile Red (cat# N1142, ThermoFisher, Waltham, MA) staining for lipid droplet accumulation was performed using well-established protocols (Oil Red O staining;[76] Nile Red staining were performed per the manufacturer's instructions).

**Senescence β-Galactosidase activity assays.** Human B-ALL cell lines ($10^6$ cells/well in a 12-well plate) were cultured in unconditioned, bone marrow stromal cell-, or adipocyte-conditioned media (depending on the experiment) for 24 h. After the incubation period, cells were harvested and lysates prepared. β-Galactosidase activity was determined using the Senescence β-Galactosidase Activity Assay Kit from Cell Signaling Technology (cat# 23833 S) per the manufacturer's protocol. Relative Fluorescence Units (RFU) of 4-MU were measured at an excitation wavelength of 360 nm and an emission wavelength of 465 nm using the Synergy Neo2 Hybrid Multi-Mode Plate Reader (BioTek, Winooski, VT).

**ELISA and Luminex analyses.** To determine the production of cytokines and chemokines from bone marrow stromal cells and adipocytes, we performed ELISA analysis of the supernatants to detect IL-6 (Invitrogen, cat# 88-7064-22), TNF-α (Invitrogen, cat# 88-7324-22), IP-10 (CXCL10) (ThermoFisher Scientific, cat# BMS6018TEN), and IL-1β (ThermoFisher Scientific, cat# BMS6002) levels per the manufacturers' protocols. For a more comprehensive analysis of cytokine and chemokine production by these cells, we performed a Luminex analysis using the Milliplex 32-plex cytokine/chemokine assay kit (Millipore Sigma, cat# MCYT-MAG-70K-PX32).

**Chemotherapy and αGalectin-9 antibody treatments.** Leukemia cell lines were preconditioned in RPMI1640 supplemented with 10% FBS, bone marrow stromal cell-conditioned media conditioned medium (SCM), or adipocyte-conditioned medium (ACM) for 18–24 h before drug treatment (chemotherapy alone, αGAL-9 alone, or the combination). After preconditioning, cells were cultured for 24–72 h (depending on the experimental question) with methotrexate (MTX; 50–70 nM), doxorubicin (DOX; 70 nM), vincristine (VIN; 2–7 nM), or αGAL-9 antibodies (10ug/mL, cat# 348902, BioLegend). For recovery experiment, cells were harvested after 48 h of drug treatment, washed with 1X PBS, and equivalent numbers of live cells were replated for an additional 3 days. Rebound after 3 days of replating was determined using trypan blue exclusion counting using the (TC20 automated cell counter, cat# 1450102, Bio-Rad). In experiments in which MTX was combined with αGal9 antibody treatment, B-ALL cells were preconditioned in unconditioned media, SCM, or ACM prior to the addition MTX and αGAL-9 antibodies at the concentrations outlined above.

**Flow cytometry, Annexin-V/PI, and EdU staining.** The surface expression of GAL9 (cat#348908, BioLegend) and TIM3 (cat#130-126-004, Miltenyi Biotec) were assessed on human B-ALL cell lines grown in RPMI, SCM or ACM media for 72 h by flow cytometry. These antibodies were also used to determine the levels of GAL-9 and TIM-3 on primary B-ALL samples and those isolated from lean and pediatric patients with obesity for all experiments including those conducted with recombinant TNF-α treatment (1–100 ng/mL, cat # 300-01 A, PeproTech). For surface stains, all antibodies were used at 1: 200 dilution.

Annexin-V-FITC/PI staining was performed to measure apoptosis per the manufacturer's protocol (cat# BMS500FI-300, eBioscience). For GFP-expressing mB-ALL cells, Annexin-V-APC/PI staining was used (cat# 640919, Biolegend).

Cell cycle analysis was performed using the Click-iT EdU AlexaFluor 488 Flow Cytometry Kit (cat# C10425, Invitrogen) per the manufacturer's instructions. After EdU labeling, 5 μL of Propidium iodide (20 μg/mL) was used to stain for DNA content in each sample. All flow samples were acquired using the BD LSR II flow cytometer (BD Biosciences) and analyzed by using Flowjo software (Ashland, Oregon).

**Western blot and cPARP experiments.** Bone marrow stromal cells and adipocyte lysates, were prepared following a method previously established[77]. Western blot analysis was performed to detect the adipocyte marker, fatty acid-binding protein-4 (FABP4, 1:4000, cat# AF1443, R&D/anti-goat HRP secondary antibody, cat#

AP180p, 1:10000, EMD Millipore). Human B-ALL cell lysates were prepared following the same protocol[77] and lysates were probed for GAL9 (cat# ab69630, 1:500, Abcam), TIM3(cat# ab47997, 1:1000, Abcam), and β-actin (cat# 4967, 1:5000, cell signaling). To assess cell cycle regulators, human B-ALL cell lysates were probed using anti-human CDK4 (cat# 12790 s, 1:500, Cell Signaling), Cyclin D3 (cat# 2936 s, 1:500, Cell Signaling), Cyclin A (cat# sc-271682, 1:500, Santa Cruz Biotechnology), Cyclin D2 (cat# 2978 S, 1:500, Cell Signaling Technology), and E2F1 (cat# sc-56661, 1:500, Santa Cruz Biotechnology). DNA damage in human B-ALL cell lysates were assessed using anti-γH2AX (cat# 2577 s, 1:1000, Cell Signaling Technology) and apoptosis was determined using anti-caspase 3 (cat# 9664 s and 9661 s, 1:1000/each, Cell Signaling Technology). We assessed the protein levels of X-linked inhibitor of apoptosis (XIAP; cat# 2042 S, 1:500, Cell Signaling Technology), BCL-xL (cat# 2764 S, 1:500, Cell Signaling Technology), BIM (cat# 2933 S, 1:500, Cell Signaling Technology), and BAX (cat# 5023 S, 1:500, Cell Signaling Technology) to determine how unconditioned, bone marrow stromal cell, and adipocyte-conditioned medium impacted modulators of cellular survival. Similar experiments were performed for ATR (cat# 2790 S, 1:500, Cell Signaling Technology) and cRAF (cat# 9422 S, 1:500, Cell Signaling Technology) to determine how DNA damage sensors were modulated in each condition, whereas, pERK (cat# 9101 S, 1:500, Cell Signaling Technology), ERK (cat# 9102 S, 1:500, Cell Signaling Technology), pAKT (cat# 9271 S, 1:500, Cell Signaling Technology) and AKT (cat# 4691 S, 1:500, Cell Signaling Technology) were analyzed as proxies for proliferation/metabolism. Either anti-mouse HRP (cat# AP308P, 1:10,000, EMD Milipore) or anti-rabbit HRP (cat# 401315, 1:10,000, Calbiochem) were used as the secondary antibody. For controls, β-actin (cat# 4967, 1:5000, Cell Signaling Technology) or β-tubulin (cat# 2144 S, 1:5000, Cell Signaling Technology) were used for all experiments. Protein signals were detected using the Bio-Rad Clarity Western ECL (cat# 170-5060, Bio-Rad), and x-ray film was developed using the Konica Minolta imager (model no. SRX-101A). Protein quantification via signal intensity determination was performed using ImageJ software 1.53n (NIH).

The detection of human B-ALL cells with cleaved PARP activation was determined via flow cytometric analysis using anti-mouse/human cleaved PARP (Asp214) conjugated to PE (cat# 552933, BD Biosciences). All flow samples were acquired using the BD LSR II flow cytometer (BD Biosciences) and analyzed by using Flowjo software (Ashland, Oregon).

**Quantitative PCR.** RNA was prepared from human B-ALL cells cultured for 3 days in RPMI, SCM, or ACM per the manufacturer's instructions (RNeasy mini kit, cat# 74104). After isolation, 1 μg of total RNA was used to synthesize cDNA using the Transcriptor First Strand cDNA Synthesis kit (cat# 04 379 012 001, Roche). cDNA was diluted to 1:10 and 2 μL of diluted cDNA was used to amplify genes of interest (see Tables 1–2 and Supplementary Table 1) by using iTaq universal SYBR Green Supermix (cat# 172–5122, Bio-Rad) to quantify gene expression. GAPDH was used as an internal control.

**Immunofluorescence.** 0.1% Poly-L-lysine (cat# P8920, Sigma-Aldrich) was added to each well of a μ-Slide 8 well-chambered coverslip (cat# 80826, Ibidi) for 2 h at 4 °C. The μ-Slide 8 well-chambered coverslip was washed with 1X PBS without Mg²⁺ and Ca²⁺ (cat# SH30378.02, HyClone). B-ALL cells were cultured for 3 days in 10% RPMI1640, SCM, or ACM in the Poly-L-lysine coated μ-Slide 8 well-chambered coverslips. Cells were fixed using 4% Paraformaldehyde in PBS (cat# J91899, Alfa Aesar), and permeabilized by adding 0.1% NP-40 (cat# ab142227, Abcam). Three washes were conducted using 1X PBS without Mg²⁺ and Ca²⁺. Cells were then incubated at room temperature in 10% Normal Goat Serum (cat# 50062Z, Life Technologies) to block nonspecific binding sites. Human B-ALL cell lines were stained overnight with the primary antibody against Gal9 (cat# ab69630, Abcam, 1:500 dilution), Tim-3 (cat# ab47997, Abcam, 1:50 dilution), and CD20 Ab-1 (cat# MS-340-S0, Thermo Fisher Scientific, 1:200 dilution). After 24 h of primary antibody incubation, cells were incubated with secondary antibodies (Goat anti-Rabbit IgG (H + L) Highly Cross-Absorbed Secondary Antibody AlexaFluor-488, cat# A11034, Invitrogen/Donkey anti-Goat IgG (H + L) Highly Cross-Absorbed Secondary Antibody AlexaFluor-568, cat# A11057, Invitrogen/Goat anti-Mouse IgG (H + L) Highly Cross-Absorbed Secondary Antibody AlexaFluor-635, cat# A31574, Invitrogen) each at a 1:100 dilution. Cells were then washed with 1X PBS, followed by 3 washes with 1X Tris Buffer Saline (TBS). After washing, cells were mounted with ProLong Gold mounting media containing DAPI (cat# P36941, Invitrogen). Images were captured using the Olympus FV1000 microscope.

**RNA-sequencing.** RNA was purified for each sample using the RNeasy Mini Kit (Qiagen) and 200–500 ng of total RNA was used as input for the Stranded mRNA-Seq kit with PolyA capture beads (cat# KK8420, KAPA Biosystems) to generate RNA-seq libraries according to the manufacturer's instructions. Final libraries were quality checked on a Bioanalyzer (Agilent) and sequenced on a NextSeq500 using PE75 chemistry at the NYU Genome Technology Center.

**Read alignments, gene sets, and PCR duplicates.** With an experimental design that employed single replicates, the typical differential expression methods to estimate statistical significance (such as p-value or FDR) were meaningless. Thus,

we used the usual practice of analyzing fold changes of genes in a pairwise manner and integrated our results by intersection approaches.

Here, in the absence of replicates, gene expression changes were determined among the phenotypic groups using a published non-parametric statistical method, Gene Integrated Sample Profile Analysis (GISPA; Kowalski et al., Nucleic Acid Research). This method is designed for three or more group comparisons with as few as one replicate per group. GISPA produces ranked gene sets within the context of an a priori specified molecular profile such as genes with increased expression specific to a group or phenotype. In the GISPA method, gene sets are formed by applying a change point model (CPM) that is defined by successive differences in variances in the distribution of transformed between-genes, within-feature profile percentiles. Because of the interpretation of these profile statistics, gene sets are ranked according to those that most satisfy the profile (change point 1), next most (change point 2) and so forth. Hence, for each comparison performed, gene set in the topmost change points (1 and 2), supporting the increased (up-regulated) and decreased (down-regulated) expression profiles in the sample group of interest (when compared to the other two groups) within each cell line were selected. Additionally, we can estimate the statistical significance of gene sets in characterizing a phenotype (or group) by gene randomization; however, such tests would not be appropriate to apply on data without replicates. We know that this is the biggest limitation of our study and we tried to mitigate the false discovery rate using non-parametric methods applicable to such experimental design with added support to the identified genes through experimental testing and validation via qPCR analysis.

Additional Notes: The data were mapped to the hg38 version of the human genome using STAR with the default parameters. Only uniquely mapped reads were used for additional analysis. GISPA was used to identify the gene-sets within the phenotypic cell lines. Gene sets were identified using the multiple change-point model (CPM) to successive differences in -log10 transformed within Feature Profile Statistics for the gene expression data using the 'change point' Bioconductor package. In the absence of biological replicates per group, we considered genes with reasonable coverage (CPM > 0.1) in all samples. Due to our low sample size, we also adjusted our p-value threshold to 0.01 to increase the stringency of our results. PCR duplicates were marked using the MarkDuplicates function in PICARD.

A Bonferroni correction test was used to validate that the significance of *LGALS9* (the gene encoding Galectin-9). Other candidates tested included ST3GAL6, LGALSL, LGALS1, LGALS3BP, LGALS8, CD44, CD69, ICAM1, ICAM2, ICAM3, ICAM4, ICAM5, PCDH1, PCDHGC3, PCDH9, CDH24, CDH2, ITGA4, and ITGB1BP1.

**Differential gene expression analysis.** The raw count expression data were normalized and rlog transformed using DESeq2[78]. Genes with zero or low expression count were filtered out prior to normalization. There may be a slight chance to filter out potential lowly expressed genes with low expression in one condition and negligible expression in other two conditions; however, in the absence of biological replicates per group, we considered genes with reasonable coverage (CPM > 0.1) in all samples (n = 27). Gene expression changes were determined among the phenotypic groups following GISPA method[79,80], which is designed for multiple group comparisons with as few as single sample per group. For each comparison, under the default settings, gene set in the topmost change points one and two, supporting the increased (up-regulated) and decreased (down-regulated) expression profiles in the sample group of interest (when compared to the two groups) within each cell line were selected. Candidate genes characterizing the groups of interest (ACM and ACM + MTX) in B-ALL cell lines are displayed using unsupervised hierarchical clustering with 1 minus pearson correlation distance and average linkage method in R heatmap.2 function. PCA plot from expression matrix and and dot plot of enriched GO sets were generated using custom scripts in R.

**External data validation analysis.** All clinical and mRNA-seq expression data for Acute Lymphoblastic Leukemia (ALL) project was downloaded from the TARGET Data Matrix Portal (TARGET Data Matrix Portal: https://ocg.cancer.gov/programs/target/data-matrix; NIH and NCI; accessed in June 2017). Phase II is comprised of tumors from pediatric patients, most who experienced an early bone marrow relapse within 4 years of initial diagnosis. Both primary (n = 137) and recurrent (n = 65) pediatric bone marrow or peripheral blood tumors collected at diagnosis with available clinical and Level 3 expression raw counts from British Columbia Cancer Agency (BCCA) sequencing center were obtained. The data was DESeq2 normalized and log2 transformed[81]. The ensemble genes were annotated using the GenCode v19 reference annotation. To analyze the effect of each individual gene in the recurrent cohort on clinical outcome, we used the median (50th percentile) based cut point to separate the patients into high expression vs low expression. Kaplan Meier Survival curves were used to test the association of the *LGALS9* expression categories and Event Free Survival (EFS). The events of interest were relapse, death due to any cause, induction failure, progression or SMN. A log-rank test was used to compare the two expression groups and the median survival time for each of the groups was reported. Similar analysis was conducted for *HAVCR2*. Survival Analysis was performed using CASAS software[82].

**shGal9 transduction (From Sigma, SHCLNG-NM_002308)**. Galectin-9 knockdown in human B-ALL cell lines was achieved using the following protocol: 1ug of target DNA (pLKO.1-shGal9) was mixed with 0.25 ug pMD2-G (envelope DNA) and 0.75ug psPAX2 packaging plasmid. Plasmids were mixed with Fugene 6 and incubated for 20 min at room temperature. DNA and lipid complexes were added to HEK293 cells, and media was changed after 24 h of transfection. Viral particles were collected at 24 and 48 h. Viral particles were pooled and used to transduce human B-ALL cells. Briefly, 6-well plates were coated with retronectin (Takara Bio, cat.# T100A/B) overnight at 4 °C. Plates were then removed and washed with PBS before adding B-ALL cells. Empty vector or viral particles were added to each well and spin-infected for 2 h at 1200 rpm (274 G). Proteins were collected after 72 h postinfection, and western blot analysis was performed to validated GAL-9 knockdown. The shRNA information is included below:

pLKO.1-puro based lenti-viral vector
443 sequence: TRCN0000057443
CCGGCGGACTTCAGATCACTGTCAACTCGAGTTGACAGTGATCT
GAAGTCCGTTTTTG
847 sequence: TRCN0000381847
GTACCGGTGGTCAGCACCTGTTTGAATACTCGAGTATTCAAA
CAGGTGCTGACCATTTTTTG

**Human Samples**. De-identified diagnosis-obtained cryopreserved primary B-ALL cells were obtained for 26 pediatric patients through the Aflac Cancer and Blood Disorders Center Leukemia and Lymphoma Biorepository (protocol number IRB00034535). De-identified age- and sex-matched healthy cryopreserved PBMCs were obtained from the Children's Healthcare of Atlanta and Emory University's Children's Clinical and Translational Discovery Core (protocol number IRB00089506). Informed consent for the inclusion of samples in the biorepositories are required prior to banking and distribution. The use of human samples for this study received an IRB exemption from the Emory University Institutional Review Board given that our usage of de-identified samples does not meet the definition of research with "human subjects" or "clinical investigation" as set forth in Emory policies and procedures and federal rules.

**Statistics**. Student's t-Test and ANOVA tests were performed when comparing 2 and 3 groups, respectively. Significance for survival and correlation analyses was determined using Log-rank and Pearson's tests, respectively. All statistical analyses were performed using graph pad prism software (version 9.1.1).

**Reporting Summary**. Further information on research design is available in the Nature Research Reporting Summary linked to this article.

## Data availability statement
The datasets generated during and/or analyzed in the current study are available from the corresponding author on reasonable request (for Table 1 and Fig. 4). The raw data and processed RNA-seq data used in this study are deposited in the GEO repository under accession number GSE183062 and can be accessed without restrictions (https://www.ncbi.nlm.nih.gov/geo/query/acc.cgi?acc=GSE183062). Source data are provided with this paper. All clinical and mRNA-seq expression data for Acute Lymphoblastic Leukemia (ALL) Phase II project was downloaded from the TARGET Data Matrix Portal (TARGET Data Matrix Portal: https://ocg.cancer.gov/programs/target/data-matrix; NIH and NCI; accessed in June 2017). Patient derived LGALS9 (GAL-9) and HAVCR2 (TIM-3) gene expression levels were determined using the St. Jude Cloud Pecan Bioportal (https://pecan.stjude.cloud/). The expression of LGALS9 and HAVCR2 in normal B-cells was determined using The Genotype-Tissue Expression (GTEx) project - GTEx Portal (https://www.gtexportal.org/home/). Source data are provided with this paper.

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

## Acknowledgements

These studies were supported by grants from the Mark Foundation for Cancer Research (Grant No. 18-031-ASP), American Cancer Society (Grant No. 00074211), Emory University Department of Pediatrics Integrated Cellular Imaging Core (Grant No. 7081000032), and Emory University School of Medicine Children's Pediatric Research Trust to C.J. Henry. Funding provided by the Transdisciplinary Research on Energetics and Cancer (TREC; Grant No. R25CA203650) also supported these studies. Primary patient samples were provided by the Aflac Cancer and Blood Disorders Center Leukemia and Lymphoma Biorepository at Children's Healthcare of Atlanta. Healthy donor PBMCs were provided by Children's Healthcare of Atlanta and Emory University's Children's Clinical and Translational Discovery Core. Research reported in this publication was supported by the Biostatistics Core, the Integrated Cellular Imaging Microscopy Core, the Integrated Computational Core, and the Integrated Proteomics Core of the Winship Cancer Institute of Emory University and NIH/NCI under award number P30CA138292. Of note, Drs. Pritha Bagchi and Jessica Randall from the Emory Integrated Proteomics Core provided invaluable assistance with BioID2 Mass Spectrometry. The content is solely the responsibility of the authors and does not necessarily represent the official views of the National Institutes of Health. All graphical images included in this manuscript were created using BioRender.com trough a fully licensed subscription (user: Jamie A.G. Hamilton). We would like to thank the authors of this manuscript and Mrs. Adeiye A. Henry for her technical assistance with experiments performed in our studies.

## Author contributions

M.L. contributed to the study design, collected/interpreted data, created manuscript figures, and assisted with drafting the Materials and Methods section of the manuscript. J.A.G.H. performed the microscopy work presented in this manuscript, mined publicly available databases for *LGALS9* and *HAVCR2* gene expression profiles in primary samples, and assisted with drafting the Materials and Methods section of the manuscript. G.R.T. assisted with the in vivo studies and data analysis. L.M. collected/interpreted data regarding murine B-ALL responses to chemotherapies. A.J.R. assisted with the primary patient studies by analyzing the surface expression of GAL-9 and TIM-3 on B-ALL cells. M.R. and B.D. provided bioinformatic analysis of the RNA sequencing data and created figures (survival analysis, heatmaps, and GO pathway analyses). S.R. provided intellectual

discussions and valuable critiques in the drafting of the manuscript. C.D.S. performed the RNA sequencing analysis and provided bioinformatic analysis of the data (PCA maps) using materials provided by J.B. D.K.G. D.D. and C.C.P. assisted with study design, technical assistance, and preparation of this manuscript. C.J.H. conceived the study, provided overview for the study design, provided financial support for the study, analyzed data, assisted with figure creation, and drafted the manuscript. All authors read and approved the final manuscript.

## Competing interests

The authors declare no competing interests.
