## [Peer Review File · Nature Communications]

Obesity-induced Galectin-9 is a Therapeutic Target in B-cell Acute Lymphoblastic LeukemiaReviewers' comments:

Reviewer #1 (Remarks to the Author); expert in galectins and immune cells:

This paper dissects the mechanism of adipose tissue mediated leukemia progression. The authors here depict a mechanism by which adipose cell-derived factors worsen disease progression along with increased chemoresistance in B-ALL. The discoveries made in this paper have the potential to augment current therapies and improve survival of B-ALL patients.

The paper has a few minor comments listed below:

1. On page 3 on the Results section, the author states that B-ALL development is potentiated by obesity, but it is disease progression and not development as the cancer cells are being injected into the animals. It is important to distinguish as the initial stages of disease development is bypassed in this experiment. The authors should change the verbiage, so it does not take away any value from the finding.
2. In figure 2, the authors depict increased survival signaling, though it would be exciting to observe cell survival over time.
3. On page 4 in Results section, the claim that "major signaling pathways are altered" is premature, as alterations in gene expression do not correspond to protein levels, which are known to undergo post-translational alterations and trigger its active state. The current study does not investigate protein levels or activation status of these proteins.
4. In page 4 under "Adipocyte Promotes Chemoresistance", the difference between ACM exposed B-ALL and ACM conditioning is not explained. The extended figure doesn't address this either. This would be an interesting distinction to observe given that the authors indicate that ACM exposed B-ALL cells perturb gene expression.
5. The RNA sequencing experiments have the potential to hint at the pathways in play here to impart resistance potentiated by adipocytes. However, the author has not focused on it or validated the targets. It might be informative for others to know these pathways and even for the current study to perhaps discuss them further in the discussion section of the paper.
6. Their drug of choice is methotrexate but in the extended data we see that ACM mediates resistance to multiple chemotherapeutics. This is an important piece of data that needs to be placed in the main article. B-ALL is generally treated with multiple drugs and therefore this piece of data makes the study stronger.
7. Extended figure 4 a and b is also another piece of data that might be better suited for the main article as it depicts that adipocytes promote production of Gal-9 in the B-ALL cells. The Gal-9 source is not clearly stated until this piece of data and the importance of adipose tissue-derived expression of Gal-9.

Overall this paper is poised to drive the field into defining the molecular basis of obesity dependent leukemia production and importantly it identifies Gal-9 as a major player in this process. Furthermore, this study is of clinical importance as it more than just identifying molecular players it proves that targeting Gal-9 in B-ALL improves prognosis, a finding that has potential to be utilized in the clinic.

Reviewer #2 (Remarks to the Author); expert in leukemia and obese mouse models:

In this paper by Lee et al, the authors analyze the direct impact of the obese environment on B-acute lymphoblastic leukemia (B-ALL) cells. They confirm previous published results that demonstrate an increased mortality in obese mice challenged with B-ALL. Using murine bone marrow stromal cell-derived adipocytes and adipocyte-conditioned medium (ACM), they show that adipocyte-secreted factors directly affects human B-ALL cell lines in vitro, leading to increased cellular aggregation and more importantly enhanced chemotherapy resistance. They identify Galectin-9 (Gal-9) as a potential candidate to explain these phenotypes. They demonstrate that adipocyte-secreted proteins up-regulate cell surface expression of Gal-9 on human B-ALL cell lines and show that antibody-mediated targeting of Gal-9 could reverse the human B-ALL characteristics acquired in the adipocytic environment. Consistent with these in vitro results, they show that targeting Gal-9 with antibody treatment reduces mouse B-ALL development in obese mice.

Specific Concerns:

1/ The manuscript presents an interesting set of observations but lacks some mechanistic insights that could be addressed in the human B-ALL cell lines.

- The authors show that ACM affects human B-ALL cell lines promoting cellular aggregation, increased expression of Gal-9 and enhanced chemotherapy resistance. They indicate that adipocyte-secreted proteins are responsible for these effects but make no efforts to determine the factor(s) in play. The use of human B-ALL cell lines and cell surface Gal-9 detection as a read-out should provide an easy screening tool for cytokine, chemokine and adipokine (...) candidates and decipher the mechanisms in play in their system.

- The authors show that adipocyte-secreted factors alter the phenotype of human cell lines. Beyond clustering, the impact of ACM (alone) on proliferation and apoptosis (if any) should be indicated. Molecularly, the authors indicate that ACM induces gene expression changes. The changes shown in Figure 2d, e, f, g are extremely modest (<2 fold) and it is unclear whether they are physiologically relevant. The use of cell lines should allow for a description of the affected signaling pathways at the protein/phosphoprotein level. Similarly a better description of the results obtained in the RNAseq experiments and some independent validations of the modified pathways would reinforce author's conclusions.

- The use of cell lines should allow for a better demonstration of Gal-9 function and a better description of its mode of action. How does recombinant Gal-9 affect B-ALL cells and their chemoresistance? The authors suggest an interaction with Tim3. Signaling pathway analysis should demonstrate this point. Finally genetic targeting of Gal9 and Tim3 in B-ALL cell lines (through ShRNA knockdown or CRISPR/cas9) should be performed to fully support authors' conclusions.

2/ Cell surface level of Gal-9 on mB-ALL cells should be determined in the in vitro experiments with unconditioned medium, SCM, and ACM (extended Figure 6) and in vivo in mB-ALL developing in lean and obese mice (Figure 6). As indicated in Figure 6a, leukemia detection in PB should be provided. It is unclear why moribund mice show difference in BM leukemic burden depending on treatment (Figure 6d).

Note: Reference for the Gal-9 neutralizing antibody used for the mouse experiments should be provided.

Authors should also determine whether Gal-9 cell expression acquired in an obese environment is maintained when cells are transferred in a normal environment. Ultimately, Gal-9 inhibition through neutralizing antibody in Xenogeneic Model with primary human B-ALL samples will dramatically increase the impact of the study (as previously described for AML by Kikushige et al, Cell Stem Cell 2015).

Other concerns:

1/ The survival profiles of lean and obese mice challenged with mB-ALL appear different in Figure 1c and Figure 6b,c. In Figure 1, 40% of lean mice survive the challenge while the other 60% mice do not show obvious changes in disease progression (kinetics). Figure 6 (vehicle) shows 0% surviving mice but a mark delay in the disease onset in the lean mice. Such apparent discrepancy should be explained (changes in methods, in mB-ALL ...??)

2/ The number of pediatric samples from lean and obese patient (Figure 4g, h, I) should be indicated. Characteristics of each cohort (Age, Sex...) should be presented. Criteria used to define lean versus obese patients should be indicated (BMI?). The authors could indicate whether there is a direct correlation between BMI and GAL-9 expression (gene expression, MFI or % of positive cells) in this obese cohort?

3/ In Figure 4d, confocal representative pictures should be changed to show individual of DAPI, CD20, Galectin-9 and Tim-3 staining as well as the "merged" picture.

Reviewer #3 (Remarks to the Author); expert in adipocytes and cancer:

The manuscript "Obesity-induced Galectin-9 is a Therapeutic Target in B-ALL" explores the role of the obese microenvironment in the pathogenesis of B-ALL. In this study, the authors show that proteins made by adipocytes can upregulate GAL-9 expression on B-ALL cells, resulting in chemoresistance. Antibody-mediated targeting of GAL-9 leads to increased apoptosis of B-ALL cells in vitro, and extends the survival of mice with this disease. Given that B-ALL is the most common childhood leukemia and given the increasing prevalence of overweight and obesity in children and adults, this study has significant biomedical relevance. However, the paper has several significant flaws that limits the impact of this work.

Major Criticism:

1. This paper is lacking a clear logical flow, which makes it difficult to link the findings into a coherent story. The following are specific issues which need to be clarified:

- a) The premise established in the introduced is that B-ALL is a prevalent childhood cancer. However, in Figure 1, the authors then study the disease in adult mice. I realize it may not be possible to model dietary obesity in juvenile mice, but this issue is never even addressed in the text.
- b) The authors also don't make it clear whether they think the adverse effects of adipocytes are mediated by marrow adipocytes or by adipocytes in peripheral fat depots. Which of these do they feel is relevant and why? Leukemia initiates in the bone marrow, but then enters the peripheral bloodstream, so one could imagine both local marrow effects and systemic effects.
- c) If the authors are interested in marrow adipocytes, they should do studies in adipocytes isolated from bone marrow from lean and obese animals. This would greatly strengthen the conclusions made here from a somewhat artificial in vitro system.

2. Figure 4 presents data from lean and obese patients. The authors should provide a demographic table that details numbers, age, gender, BMI, and comorbidities. They may also consider performing a regression analysis to identify key variables associated with their outcome of interest.

3. The studies with the GAL-9 antibodies have several important limitations:

- a) The authors never show data to confirm that the antibody used specifically targets GAL-9.
- b) If the antibody does in fact target GAL-9 is it acting as a neutralizing antibody or could it be acting as an activating antibody? Does antibody action phenocopy GAL-9 loss of function through genetic means?
- c) Where else is GAL-9 expressed and what effects might the antibody binding at these sites have?

Minor Criticism:

1. In Figure 1, rather than a photograph and final weights, the authors should show weight curves.

2. For the studies in Figure 2 and elsewhere, it is not clear to me that unconditioned media and stromal conditioned media are the most appropriate controls. The authors should think about using conditioned media from another differentiated cell type.

3. Figure 2B shows an increase in IL-6 and TNF α in adipocyte-conditioned media, but the authors then never come back to this. What protein(s) in adipocyte-conditioned media do they think are mediating the upregulation of GAL-9. Are IL-6 and/or TNF α involved?

4. The authors need to clarify the statistics used in Figure 4B and 4C. The error bars overlap in most cases, yet many of the data are listed as showing statistically significant differences.

5. The studies in Figure 5 on the cell cycle and apoptosis would be strengthened by showing a secondary assay such as cleaved caspase-3.

6. The legend for extended data Figure 1 describes a panel E, but the figure only goes up to panel D.

Reviewer 1

Minor Comments:

1. On page 3 of the Results section, the author states that B-ALL development is potentiated by obesity, but it is disease progression and not development as the cancer cells are being injected into the animals. It is important to distinguish as the initial stages of disease development is bypassed in this experiment. The authors should change the verbiage, so it does not take away any value from the finding.

This was an excellent observation and the verbiage is now changed to B-ALL development.

2. In figure 2, the authors depict increased survival signaling, though it would be exciting to observe cell survival over time. **This is an excellent point. We have now added Figure 2b in the revised manuscript and this depicts cell survival over time. Interestingly, we found the B-ALL cells appear to be growing slower when cultured in adipocyte-conditioned media (despite the maintenance or increase in genes regulating proliferation and survival). We have found that reduced B-ALL cell cycle progression in ACM is mediated by Galectin-9, which is a major observation of this study.**

3. On page 4 in Results section, the claim that “major signaling pathways are altered” is premature, as alterations in gene expression do not correspond to protein levels, which are known to undergo post-translational alterations and trigger its active state. The current study does not investigate protein levels or activation status of these proteins. **This comment was very insightful, and Reviewer 1 is completely correct. To more accurately reflect our findings, the verbiage in the revised manuscript is now restated as “changes in gene expression profiles”.**

4. In page 4 under “Adipocyte Promotes Chemoresistance”, the difference between ACM exposed B-ALL and ACM conditioning is not explained. The extended figure doesn’t address this either. This would be an interesting distinction to observe given that the authors indicate that ACM exposed B-ALL cells perturb gene expression. **The verbiage “ACM exposed B-ALL” and “ACM conditioning” are interchangeable; however, it was confusing as presented in the original submission. To clarify our experimental approach, we have now modified the language in this section, which now states, “we next sought to determine how pre-conditioning B-ALL cells with ACM impacted their response to chemotherapy treatment”. This revision is found on line 97 of the resubmitted manuscript.**

5. The RNA sequencing experiments have the potential to hint at the pathways in play here to impart resistance potentiated by adipocytes. However, the author has not focused on it or validated the targets. It might be informative for others to know these pathways and even for the current study to perhaps discuss them further in the discussion section of the paper. **This was an excellent point, which we have addressed in the revised manuscript. We have dedicated an entire section of the manuscript to exploring in greater detail how gene expression changes impart chemoresistance to human B-ALL cells (page 5; lines 127-148). Importantly, we have also clarified how this experiment resulted in the discovery of adipocyte-induced, Galectin-9 mediated chemoresistance. As suggested, we have also dedicated an entire figure demonstrating our attempts to validate gene targets nominated by our RNA-sequencing study (Extended Data Fig. 5). Furthermore, a new section that we have included in our discussion now highlights potential mechanisms of adipocyte-induced chemoresistance in human B-ALL cells nominated by our RNA-sequencing results (page 8; lines 283-291).**

6. Their drug of choice is methotrexate but in the extended data we see that ACM mediates resistance to multiple chemotherapeutics. This is an important piece of data that needs to be placed in the main article. B-ALL is generally treated with multiple drugs and therefore this piece of data makes the study stronger. **This was an excellent recommendation. The revised Figure 3 now shows representative primary data for how REH cells respond to methotrexate, doxorubicin, and vincristine in unconditioned medium, stromal cell-conditioned medium, and adipocyte-conditioned medium (Fig. 3a). Additionally, the quantitative data is shown for 3 human B-ALL cell lines treated with the chemotherapies stated above (Fig. 3b-d).**

7. Extended figure 4 a and b is also another piece of data that might be better suited for the main article as it depicts that adipocytes promote production of Gal-9 in the B-ALL cells. The Gal-9 source is not clearly stated until this piece of data and the importance of adipose tissue-derived expression of Gal-9. **This was also an excellent recommendation. Extended Figure 4a in the original manuscript is now included in Figure 5 of the main article. Furthermore, we have modified our language throughout the revised manuscript to reinforce our finding that GAL-9 is being upregulated by B-ALL cells as a cell intrinsic response to adipocyte exposure.**

Reviewer 2

Specific Concerns:

1/ The manuscript presents an interesting set of observations but lacks some mechanistic insights that could be addressed in the human B-ALL cell lines.

- The authors show that ACM affects human B-ALL cell lines promoting cellular aggregation, increased expression of Gal-9 and enhanced chemotherapy resistance. They indicate that adipocyte-secreted proteins are responsible for these effects but make no efforts to determine the factor(s) in play. The use of human B-ALL cell lines and cell surface Gal-9 detection as a read-out should provide an easy screening tool

for cytokine, chemokine and adipokine (...) candidates and decipher the mechanisms in play in their system. **This was an excellent point which has been addressed in the revised manuscript. After testing how various adipocyte-secreted cytokines and chemokines impact B-ALL function, we found that TNF- α treatment of multiple human B-ALL cell lines induced significant Galectin-9 surface expression (Fig. 5d). This observation corroborates similar observations in astrocytes and synovial fibroblasts, where TNF- α , IFN- γ , TLR3 agonist, and TLR4 agonist stimulation upregulates Galectin-9 surface expression (Steelman *et al.*, The Journal of Biological Chemistry, 2013/Pearson *et al.*, Scientific Reports, 2018).**

- The authors show that adipocyte-secreted factors alter the phenotype of human cell lines. Beyond clustering, the impact of ACM (alone) on proliferation and apoptosis (if any) should be indicated. Molecularly, the authors indicate that ACM induces gene expression changes. The changes shown in Figure 2d, e, f, g are extremely modest (<2 fold) and it is unclear whether they are physiologically relevant. The use of cell lines should allow for a description of the affected signaling pathways at the protein/phosphoprotein level. Similarly a better description of the results obtained in the RNAseq experiments and some independent validations of the modified pathways would reinforce author's conclusions. **In our revised manuscript, we now include data describing the impact of unconditioned media, stromal cell-conditioned medium (SCM), and adipocyte-conditioned medium (ACM) on B-ALL cell proliferation and apoptosis. Surprisingly, we found that culturing human B-ALL cells resulted in slower proliferation *in vitro* (new Fig. 2b) which was not accompanied by significant cell death in most of the B-ALL cell lines tested (new Fig. 2c).**

On page 5 (lines 127-148) we now have included a separate section in which we discuss the results of the RNA-sequencing experiment in more detail. Additionally, we have added a new figure to the revised manuscript in which we attempt to validate (via qPCR analysis) candidate genes nominated by our RNA-sequencing experiment (Extended Data Fig. 5). Furthermore, we now explain how this experiment nominated Galectin-9 for further study and how this lectin became the focal point for further experimentation. In the Discussion section of our revised manuscript, we now go into more detail describing potential mechanisms of adipocyte-induced chemoresistance in human B-ALL cells nominated by our RNA-sequencing results (page 8; lines 283-291).

We have also performed proteomic analysis of multiple human B-ALL cells cultured in unconditioned media, SCM, and ACM in the presence and absence of methotrexate, as well as, proteomic analysis of serum samples taken from lean and obese patients with B-ALL. Unfortunately, the analysis of this data is incomplete at the time of this resubmission and we are still awaiting the results from the serum analysis. In future studies, we would like to integrate our proteomics data with metabolomics data we have recently obtained to focus on a more "omics"-centric manuscript detailing the impact of adiposity on B-ALL progression. Given the incomplete nature of our proteomics data, we feel that a more thorough analysis is required prior to publication consideration.

- The use of cell lines should allow for a better demonstration of Gal-9 function and a better description of its mode of action. How does recombinant Gal-9 affect B-ALL cells and their chemoresistance? The authors suggest an interaction with Tim3. Signaling pathway analysis should demonstrate this point. Finally genetic targeting of Gal9 and Tim3 in B-ALL cell lines (through ShRNA knockdown or CRISPR/cas9) should be performed to fully support authors' conclusions. **These are all excellent critiques. To this end, we have found that treating human B-ALL cell lines with recombinant Galectin-9 does not induce cytotoxicity in the presence or absence of methotrexate (Extended Data Fig. 8g-h). This is an important observation because it demonstrates that inhibiting Galectin-9 surface activity promotes B-ALL cell death (α GAL-9 antibody *in vitro* studies); whereas, soluble Galectin-9 is incapable of**

inducing apoptosis. Regarding the specificity of the α GAL-9 antibody, we found that α GAL-9 antibody mediated cytotoxicity was significantly attenuated when Galectin-9 proteins levels were reduced via genetic targeting (ShRNA knockdown; Extended Figure 8f). In Extended Data Fig. 7, we report that TIM3 gene expression and surface expression are not impacted, either directly or indirectly, by adiposity. However, we observed increased co-localization of GAL-9 and TIM-3 in B-ALL cells cultured in ACM. At this time, we have not defined the importance of this interaction. However, we are actively studying this relationship in my laboratory and are attempting to overcome technical difficulties we have encountered with genetically targeting TIM3 through shRNA knockdown or CRISPR/cas9 methodologies.

2/ Cell surface level of Gal-9 on mB-ALL cells should be determine in the in vitro experiments with unconditioned medium, SCM, and ACM (extended Figure6) and in vivo in mB-ALL developing in lean and obese mice (Figure 6). As indicated in Figure 6a, leukemia detection in PB should be provided. It is unclear why moribund mice show difference in BM leukemic burden depending on treatment (figure 6d). Note: Reference for the Gal-9 neutralizing antibody used for the mouse experiments should be provided. **This was a great observation, and we now demonstrate that culturing mB-ALL cells in ACM results in a significant induction of Galectin-9 surface expression (Extended Data Fig. 9c-d). Furthermore, we now provide data demonstrating that transplanting mB-ALL in obese mice results in a 3-fold increase in the percentage of Galectin-9 positive cells; whereas, Gal-9 positive mB-ALL cells do not significantly expand upon transplantation into lean mice. Furthermore, additional experiments did not corroborate our initial findings of differences in treatment-dependent BM leukemic burden; therefore, these results were removed from our study. As noted above, we have included the reference for the α GAL-9 antibody used for our *in vivo* studies in the revised manuscript.**

Authors should also determine whether Gal-9 cell expression acquired in an obese environment is maintained when cells are transferred in a normal environment. Ultimately, Gal-9 inhibition through of neutralizing antibody in Xenogeneic Model with primary human B-ALL samples will dramatically increases the impact of the study (as previously described for AML by Kikushige et al, Cell Stem Cell 2015). **These are excellent points, and our experiments conducted in response to these inquiries revealed that Galectin-9 surface expression is downregulated on ACM-exposed B-ALL cells when re-cultured in unconditioned media (Fig. 5e-f). This observation demonstrates that sustained Galectin-9 surface expression on B-ALL cells requires persistent exposure to adipose-rich microenvironments. Furthermore, in xenograft experiments, we found that α GAL-9 antibody treatment conferred significant protection to obese, but not lean, mice; whereas, the converse was true for methotrexate treatment (Fig. 7d-e). These observations corroborate outcomes in wild-type mice, which were reported in the initial submission.**

Other concerns:

1/ The survival profiles of lean and obese mice challenged with mB-ALL appear different in Figure 1c and Figure 6b,c. In Figure 1, 40% of lean mice survive the challenge while the other 60% mice do not show obvious changes in disease progression (kinetics). Figure 6 (vehicle) shows 0% surviving mice but a mark delay in the disease onset in the lean mice. Such apparent discrepancy should be explained (changes in methods, in mB-ALL ...??). **To address this inquiry, differences in survival outcomes raised in this comment were as a result of varying numbers of B-ALL transplanted into mice for each experiment. In an attempt to increase transparency and to ease the readers' interpretation of our *in vivo* studies, we now report the approximate number of B-ALL cells transplanted into mice in the legends where murine experiments were conducted.**

2/ The number of pediatric samples from lean and obese patient (Figure 4g, h, I) should be indicated. Characteristics of each cohort (Age, Sex...) should be presented. Criteria used to define lean versus obese patients should be indicated (BMI?). The authors could indicate whether there is a direct correlation between BMI and GAL-9 expression (gene expression, MFI or % of positive cells) in this obese cohort? **Tables containing the requested information are now provided in the revised manuscript. We also performed linear regression to analyze BMI vs. surface GAL-9 expression, which revealed a significant correlation between these parameters (Fig 5k).**

3/ In Figure 4d, confocal representative pictures should be changed to show individual of DAPI, CD20, Galectin-9 and Tim-3 staining as well as the “merged” picture. **We sincerely appreciate Reviewer 2’s critique of our confocal images. This error was corrected, and we now present each channel and merged images in our revised manuscript (Extended Data Fig. 7b)**

Reviewer 3

Major Criticism:

1. This paper is lacking a clear logical flow, which makes it difficult to link the findings into a coherent story. The following are specific issues which need to be clarified:

Based on this critique, figures have been substantially reorganized and sections are now presented in smaller, more coherent units. In summary, the manuscript reads in 3 parts: 1. The impact of obesity on B-ALL progression and the function of B-ALL cells, 2. The identification of Galectin-9 on B-ALL cells and its functional importance, and 3. The potential of antibody-mediated targeting of Galectin-9 on B-ALL cells in obese microenvironment as a novel therapeutic option.

a) The premise established in the introduced is that B-ALL is a prevalent childhood cancer. However, in Figure 1, the authors then study the disease in adult mice. I realize it may not be possible to model dietary obesity in juvenile mice, but this issue is never even addressed in the text. **Obesity is reaching pandemic levels in pediatric and adult populations, and results in poorer survival outcomes in both demographics. We have now highlighted this relationship in both populations (lines 59-62) in the revised manuscript, which we feel increases the scope of our findings given the murine and clinical samples analyzed in our study.**

b) The authors also don’t make it clear whether they think the adverse effects of adipocytes are mediated by marrow adipocytes or by adipocytes in peripheral fat depots. Which of these do they feel is relevant and why? Leukemia initiates in the bone marrow, but then enters the peripheral bloodstream, so one could imagine both local marrow effects and systemic effects. **This is an excellent observation. To address this critique, we now provide data showing that soluble factors from primary adipocytes isolated from subcutaneous fat depots are capable of conferring chemoresistance to B-ALL cells (Extended Data Fig. 3). Given these observations, we conclude that adipocytes are capable of conferring chemoresistance to B-ALL cells regardless of their tissue of origin. Furthermore, we conclude that the ability of adipocytes to confer chemoresistance to B-ALL cells may occur through local (soluble factors and contact dependency; supported by data presented in Fig. 5c) and systemic (soluble factor mediated) mechanisms.**

c) If the authors are interested in marrow adipocytes, they should do studies in adipocytes isolated from bone marrow from lean and obese animals. This would greatly strengthen the conclusions made here from a somewhat artificial in vitro system. **This was an excellent suggestion; however, technical limitations inhibited our ability to recover a significant number of adipocytes from the bone marrow of lean or obese mice to conduct the proposed experiment. Therefore, we attempted to determine if**

adipocytes isolated from other tissue were capable of conferring chemoresistance to B-ALL cells. Indeed, we found that adipocytes obtained from subcutaneous fat depots were capable of inducing chemoresistance to methotrexate in multiple human B-ALL cell lines (Extended Data Fig. 3).

2. Figure 4 presents data from lean and obese patients. The authors should provide a demographic table that details numbers, age, gender, BMI, and comorbidities. They may also consider performing a regression analysis to identify key variables associated with their outcome of interest. **This was an excellent point, which we have addressed in the revised manuscript. Experiments with primary human samples were performed multiple times in our study, and accompanying tables are now provided. Furthermore, regression analyses were performed which revealed a significant correlation between BMI and Galectin-9 surface expression (Fig. 5k).**

3. The studies with the GAL-9 antibodies have several important limitations:
a) The authors never show data to confirm that the antibody used specifically targets GAL-9.
b) If the antibody does in fact target GAL-9 is it acting as a neutralizing antibody or could it be acting as an activating antibody? Does antibody action phenocopy GAL-9 loss of function through genetic means? **To address these questions, we used shRNA methodology and successfully knocked down GAL-9 protein expression in human B-ALL cells. We observed a significant increase in cell death when Galectin-9 KD B-ALL cells were grown in adipocyte-conditioned medium but not unconditioned medium, which phenocopied responses observed in parental cells treated with α GAL-9 antibody (Extended Data Fig. 8f). Furthermore, α GAL-9 antibody treatment failed to kill Galectin-9 KD B-ALL cells (Extended Data Fig. 8f). These results highlight the specificity of the α GAL-9 antibody and further emphasize the importance of GAL-9 in preventing the apoptosis of B-ALL cells when exposed to adipocyte-secreted factors.**

Furthermore, we found that α GAL-9 antibody treatment increase the expression of genes involved in B-ALL homeostasis (Extended Data Fig. 8c-e). The increased gene expression is consistent with increase cell cycle progression observed after α GAL-9 antibody treatment of B-ALL cells in ACM (Fig. 6). These observations suggest that the “hyperactivation” of B-ALL cells promotes cell death.

c) Where else is GAL-9 expressed and what effects might the antibody binding at these sites have? **In the revised manuscript we now provide human transcriptomic data from the BloodSpot database which reveal high gene expression levels for *LGALS9* in hematopoietic stem cells, NK cells, naïve B-cells, granulocytes, monocytes, and myeloid DCs (Extended Data Fig. 6j). The surface expression of the GAL-9 on each of these populations will need to be determined in order to adequately assess potential toxicity associated with α GAL-9 antibody treatment (although no significant toxicity issues were observed in murine studies). In the revised manuscript, we have now included a discussion of these points.**

Minor

Criticism:

1. In Figure 1, rather than a photograph and final weights, the authors should show weight curves. **To clarify, the photograph of mice (Fig. 1a) and weights (Fig. 1b) were taken prior to injecting mice with B-ALL cells. The goal of showing this data was to demonstrate to the reader the extent of obesity induced in wild-type mice prior to the injection of B-ALL cells. I have now made this distinction in the revised manuscript.**

2. For the studies in Figure 2 and elsewhere, it is not clear to me that unconditioned media and stromal conditioned media are the most appropriate controls. The authors should think about using conditioned media from another differentiated cell type. **This was an excellent suggestion made by Reviewer 3. To**

this end, we performed similar experiments using fibroblast- and osteoblast-conditioned medium, and found that they were unable to promote chemoresistance in human B-ALL cells treated with methotrexate (Extended Data Fig. 3).

3. Figure 2B shows an increase in IL-6 and TNF α in adipocyte-conditioned media, but the authors then never come back to this. What protein(s) in adipocyte-conditioned media do they think are mediating the upregulation of GAL-9. Are IL-6 and/or TNF α involved? **This was also an excellent critique, which significantly added to our understanding of Galectin-9 regulation on human B-ALL cells after exposure to adipocyte-conditioned medium. In recombinant cytokine experiments, we found that TNF- α treatment was sufficient to induce GAL-9 upregulation on human B-ALL cells. These data are now provided in the revised manuscript (Fig. 5d).**

4. The authors need to clarify the statistics used in Figure 4B and 4C. The error bars overlap in most cases, yet many of the data are listed as showing statistically significant differences. **Thank you for these important observations. We have carefully reviewed all statistical analyses in the manuscript, and corrected anything that was misstated in the initial submission.**

5. The studies in Figure 5 on the cell cycle and apoptosis would be strengthened by showing a secondary assay such as cleaved caspase-3. **This was a fantastic suggestion, and we now report that caspase-3 activation increases significantly in ACM-exposed B-ALL cells treated with α GAL-9 antibody (Fig. 6g-h).**

6. The legend for extended data Figure 1 describes a panel E, but the figure only goes up to panel D. **The original Extended Data Fig. 1 has been replaced with a new Extended Data Fig. 1 which contains the appropriate legend. Furthermore, we have carefully checked the information presented in each legend to mitigate errors. Thank you for bringing this matter to our attention.**

Reviewers' comments:

Reviewer #1 (Remarks to the Author):

Authors have adequately addressed all concerns and recommendations.

Charles J Dimitroff, PhD

Reviewer #2 (Remarks to the Author):

The authors provide an extensive revised manuscript by reshuffling and changing the presentation format of previously presented data and adding some novel piece of data. Unfortunately, important criticisms formulated in the precedent review have not been properly addressed in this revised manuscript.

These points are reiterated below:

Specific Concerns:

1/ Mechanistic insights

- Adipocyte-secreted proteins that impact human B-ALL cell lines?

The authors found that TNF- α treatment is sufficient to induce Galectin-9 surface expression on human B-ALL cell lines. It is unclear from the manuscript how many candidates have been tested. It is unclear that TNF- α is actually required for the Galectin-9 induction by adipocytes or ACM (a point that can be demonstrated by blocking TNF- α or its pathway in culture). It is also unclear whether TNF- α alone is able to mimic the impact of adipocytes or ACM on the B-ALL cell biology (cluster formation/proliferation) and chemo-resistance.

- Effects of adipocyte-secreted factors on the phenotype of human B-ALL cell lines

a/ In presence of ACM, the authors show slower cell growth with minimal increase in apoptosis. No explanation is provided for this phenotype. Meanwhile, the cell cycle analyses described in Figure 6b show minimal alterations in presence of ACM. These data should be reconciled.

b/ The interpretation of the NF- κ B, AKT, ERK1 and STAT5 gene expression levels is still unclear. Why are these genes and changes meaningful? Are these pathways supposed to be transcriptionally regulated? What is the conclusion for these data? It is concerning that the authors can positively interpret both a significant increase (original version) and the absence of significant changes (revised version).

Later in the manuscript, the authors use non-significant increased AKT and ERK1 gene expression as a marker of cell activation (Extended Data Fig. 8c-e). Even if it was statistically significant, the basis of this link is not obvious. The authors' hypothesis should be directly tested through immunoblot, IF, flow cytometry... to show objective evidences of "cell activation".

c/ The authors indicate that they are working on completing global proteomic and metabolomic studies. It is a worthwhile and certainly long-term project. However, it is unclear why this would preclude targeted immunoblot analyses on signaling pathways and cell cycle regulators on these cell lines to explain the observed phenotypes.

d/ The presentation of the results obtained in the RNAseq experiments has not been significantly improved. The difficulty in validating the described pathways also raises questions about the robustness of the results obtained by RNAseq.

Figure 4a-b: the authors indicate "Principal component analysis (PCA) and unsupervised hierarchical clustering of RNA-sequencing results revealed that global gene expression profiles were surprisingly similar in human B-ALL cell lines cultured in RPMI and SCM; however, ACM exposure induced

dramatic changes in gene expression in leukemia cells (Fig. 4a-b)." This statement does not seem to be true for the REH cell line shown in Figure 4a where RPMI, SCM and ACM conditions show minimal differences in PCA.

Figure 4e: Presentation of the data in this figure is not convincing. As presented, the figure is not readable and cannot be interpreted. Bigger font should be used. Figure with curated data that show the most relevant gene sets between different conditions (RPMI vs SCM vs ACM; + vs - MTX...) would be more convincing. Complete differential gene expression (+fold changes / p values) and GSEA data (+p values) could be presented in excel table as supplementary. Primary data should be made available in an open public repository (GEO number not indicated)

e/ Gal-9 function and a better description of its mode of action.

Overall, the authors provide minimal new information about the mode of action of Tim3/Gal9 in B-ALL. Immunoblot analyses (or other analyses) with these cell lines should have addressed this question as previously done by Kikushige et al, Cell Stem Cell 2015. The results obtained with GAL-9 knock-down are encouraging but limited to a single assay, and therefore, seem to be preliminary.

2/ "Ultimately, Gal-9 inhibition through of neutralizing antibody in Xenogeneic Model with primary human B-ALL samples will dramatically increases the impact of the study (as previously described for AML by Kikushige et al, Cell Stem Cell 2015)."

The authors validate their results on human cell using the 697 human cell line in immunodeficient NOG, which represent a less compelling model than primary human B-ALL samples, but still a noticeable result.

Other concerns:

Figure 4d: Confocal representative pictures shown in this figure are not consistent with previously results obtained by flow cytometry analysis. Compared to RPMI condition, Gal9 appears to be upregulated in SCM media in similar way than in ACM media (compared to flow panel in Figure 5a). Tim3 expression seems to be more expressed in ACM than RPMI and SCM conditions (compared to flow panel in Extended Figure 7a). These discrepancies question the robustness of these data and should be explained. The low CD20 expression in the RPMI condition is also surprising.

Reviewer #3 (Remarks to the Author):

In this revised manuscript, the authors have made significant revisions to the text and have added a number of additional experiments. Collectively, these changes have dramatically improved the manuscript and all of my points of criticism have been sufficiently addressed.

I would suggest the authors address the following minor points:

1. Line 124 has a typo and should read "are more resistant".
2. Lines 145-148 describe the nomination of Galectin 9. Beyond "further assessment" what criteria made the authors focus on this molecule?
3. The text in Figure 4e is too small to be visualized.

Reviewer #4 (Remarks to the Author):

The study is well designed, and the results are very interesting and highly relevant to the still obscure relationship between obesity and other diseases, in this case Leukaemia. The main caveat of the RNAseq part of the study is the lack of technical or biological replicates, which the authors partially

overcome by qPCR replication of the results. Nevertheless, what the qPCR replication cannot provide are the differentially expressed genes that would be identified in addition to GAL-9 due to this technical limitation and authors should acknowledge that in the discussion part of the study. Aside of that, there is some critical information lacking the methodology of RNAseq used on the side of the bioinformatic pipeline which is crucial for evaluation and needs to be included in the manuscript for reproducibility:

- 1) The authors mention nothing of the alignment to the reads to the reference. In mouse and human data which version was used mm10 and hg38? Something else? Which aligner was used, STAR, bowtie or something else? How did the authors, if in any way, deviated from the best practices/defaults of whatever aligner they used?
- 2) What statistical test and, most importantly, multiple testing correction did the authors use for the RNAseq data? FDR?
- 3) Which are the 20 candidates for adhesion molecules investigated in the RNAseq data? Which criteria was used for selecting this list and not other? In this test how was multiple testing correction applied?
- 4) The authors state that they removed from further analysis before normalisation genes with no expression or "low expressed" genes? What is the cut-off levels and how many genes did this cut-off?
- 5) How did the authors manage the PCR duplicates in the RNAseq data?
- 6) How did the authors manage the multimapping reads in the RNAseq data?

Reviewer 1

Comments:

1. Authors have adequately addressed all concerns and recommendations.

-Charles J Dimitroff, PhD

Response: We sincerely appreciate Dr. Dimitroff's feedback and thank him for his assistance in improving our manuscript.

Reviewer 2

Comments:

The authors provide an extensive revised manuscript by reshuffling and changing the presentation format of previously presented data and adding some novel piece of data. Unfortunately, important criticisms formulated in the precedent review have not been properly addressed in this revised manuscript.

These points are reiterated below:

1. Adipocyte-secreted proteins that impact human B-ALL cell lines?

The authors found that TNF- α treatment is sufficient to induce Galectin-9 surface expression on human B-ALL cell lines. It is unclear from the manuscript how many candidates have been tested. It is unclear that TNF- α is actually required for the Galectin-9 induction by adipocytes or ACM (a point that can be demonstrated by blocking TNF- α or its pathway in culture). It is also unclear whether TNF- α alone is able to mimic the impact of adipocytes or ACM on the B-ALL cell biology (cluster formation/proliferation) and chemo-resistance.

Response: We now present new data where we have tested the impact of 3 cytokines (TNF- α , IL-6, IL-1 β) and 1 chemokine (IP-10) on their ability to modulate Galectin-9 (GAL-9) surface expression on human B-ALL cells (**Fig. 5d** and **Extended Data Fig. 7a-f**). These adipocyte-secreted inflammatory mediators were chosen based on Luminex and ELISA experiments which demonstrated significant increases in their secretion by adipocytes relative to bone marrow stromal cells. Of these, we found that only recombinant TNF- α was capable of upregulating GAL-9 surface expression on human B-ALL cells. Interestingly, neutralization of TNF- α from the adipocyte secretome did not significantly decrease GAL-9 surface expression on human B-ALL cells. These observations demonstrated that TNF- α is not solely responsible for GAL-9 surface expression on B-ALL cells and that other adipocyte-secreted factors can also contribute. The identification of additional adipocyte-secreted factors which upregulate GAL-9 surface expression on B-ALL cells, as well as other leukemia and lymphoma subtypes, is an active area of investigation in my laboratory, but is beyond the scope of this manuscript in which we have focused on the identification and validation of GAL-9 as a therapeutic target in ALL in the context of obesity.

2. Effects of adipocyte-secreted factors on the phenotype of human B-ALL cell lines.

a/ *In presence of ACM, the authors show slower cell growth with minimal increase in apoptosis. No explanation is provided for this phenotype. Meanwhile, the cell cycle analyses described in Figure 6b show*

minimal alterations in presence of ACM. These data should be reconciled.

Response: We have conducted several additional experiments and molecular profiling of cells grown in unconditioned, bone marrow stromal cell, and adipocyte conditioned medium. These experiments have led to our inclusion of new figures which drastically increase our understanding of how human B-ALL cells respond to the adipocyte secretome in the absence of drug treatment (**Fig. 2d-e; Extended Data Fig. 3; Extended Data Fig. 9**).

From these new data, we have concluded that the reduction in proliferation of human B-ALL cells cultured in ACM is due to the induction of senescence. This is a major finding, which is supported by increased β -Gal positivity of human B-ALL cells when cultured in ACM and not the other conditions tested. We also provide additional molecular data (western blot analyses) which supports this conclusion, in which we show that the protein levels of inducers of senescence (AKT, ERK, BCL-xL, and XIAP) are upregulated in ACM-exposed B-ALL cells. Thus, the decreased cell numbers are not due to widespread apoptosis or significant cell cycle alteration, but cellular senescence.

***b/** The interpretation of the NF- κ B, AKT, ERK1 and STAT5 gene expression levels is still unclear. Why are these genes and changes meaningful? Are these pathways supposed to be transcriptionally regulated? What is the conclusion for these data? It is concerning that the authors can positively interpret both a significant increase (original version) and the absence of significant changes (revised version).*

Later in the manuscript, the authors use non-significant increased AKT and ERK1 gene expression as a marker of cell activation (Extended Data Fig. 8c-e). Even if it was statistically significant, the basis of this link is not obvious. The authors' hypothesis should be directly tested through immunoblot, IF, flow cytometry... to show objective evidences of "cell activation".

***c/** The authors indicate that they are working on completing global proteomic and metabolomic studies. It is a worthwhile and certainly long-term project. However, it is unclear why this would preclude targeted immunoblot analyses on signaling pathways and cell cycle regulators on these cell lines to explain the observed phenotypes.*

Responses (to b and c): We have now replaced some of our initial qPCR results with western blot analyses of gene products nominated by our RNA-sequencing and qPCR experiments. In the new **Figure 2e** we report that AKT and ERK activation is increased in ACM-cultured human B-ALL cells despite the reductions in proliferation observed in this condition. Notably, the activation of other tested proteins such as NF- κ B were equivalent in each condition. As now discussed in more detail in our revised manuscript, we conclude that the activation of these pathways in ACM-cultured human B-ALL is associated with the induction of senescence (along with other supporting experiments) which mitigates DNA-damage induced apoptosis.

In addition to performing western blot analysis of the aforementioned proteins, we also increased the number of cell cycle regulators we analyzed under each condition. In the previous submission, we demonstrated that Cyclin D3 and CDK4 protein levels were completely abolished in α GAL-9 antibody treated B-ALL cells, only when cells were cultured in ACM (**Fig. 6e-f**). We now report that this is also the case for Cyclin A and CDK2; whereas, we see increases in E2F protein levels (which promote cell cycle progression) after antibody treatment of ACM-exposed human B-ALL cells (**Extended Data Fig. 9a-d**). The original and additional data add further support to our conclusion that α GAL-9 antibody treatment of ACM-exposed human B-ALL cells disrupts the protein levels of cell cycle regulators which is associated with enhanced cell cycle progression (and death) of human B-ALL cells.

d/ The presentation of the results obtained in the RNAseq experiments has not been significantly improved. The difficulty in validating the described pathways also raises questions about the robustness of the results obtained by RNAseq.

Response: We have simplified our presentation of the RNA-seq data in the revised **Fig. 4 and Supplementary Table 1**. Moreover, we now only report on data which are validated by qPCR, western blot, or flow cytometric analyses (discussed hits had to validate using at least two of these methods). As in any gene expression dataset, we identified false positives, which did not validate in the qPCR analysis. These data are now omitted from this manuscript to avoid superfluous information. In addition, we have deposited the data in GEO for others to assess the RNA-seq data themselves, if questions remain.

Figure 4a-b: the authors indicate "Principal component analysis (PCA) and unsupervised hierarchical clustering of RNA-sequencing results revealed that global gene expression profiles were surprisingly similar in human B-ALL cell lines cultured in RPMI and SCM; however, ACM exposure induced dramatic changes in gene expression in leukemia cells (Fig. 4a-b)." This statement does not seem to be true for the REH cell line shown in Figure 4a where RPMI, SCM and ACM conditions show minimal differences in PCA.

Response: We agree that the PCA figures did not adequately demonstrate the differences in gene expression in REH cells cultured in ACM. Therefore, we have separated the medium alone from the drug-treated groups. The new **Fig. 4a-d** clearly show that the global gene expression profiles are altered by exposure to ACM, which occurs in the absence and presence of methotrexate treatment.

Figure 4e: Presentation of the data in this figure is not convincing. As presented, the figure is not readable and cannot be interpreted. Bigger font should be used. Figure with curated data that show the most relevant gene sets between different conditions (RPMI vs SCM vs ACM; + vs - MTX...) would be more convincing. Complete differential gene expression (+fold changes / p values) and GSEA data (+p values) could be presented in excel table as supplementary. Primary data should be made available in an open public repository (GEO number not indicated).

Response: We have removed the former figure 4e and simplified **Fig. 4a-d**, which now show the PCA data broken down into treatment groups for each B-ALL cell line. We have also added red and green arrows to

Fig. 4f to highlight increased and decreased genes which are associated with chemoresistance in both human B-ALL cell lines. Given the extent of the data presented in **Fig. 4f**, and to avoid presenting illegible font sizes, we now present the genes indicated by the red and green arrows in Fig. 4f in **Supplementary Table 1**. Additionally in **Table 1**, we also provide new data indicating additional targets of chemoresistance we analyzed in our study. As noted, our RNA-sequencing results have been uploaded to the GEO repository for re-analyses (GEO accession number GSE183062).

e/ GAL-9 function and a better description of its mode of action.

Overall, the authors provide minimal new information about the mode of action of Tim3/Gal9 in B-ALL. Immunoblot analyses (or other analyses) with these cell lines should have addressed this question as previously done by Kikushige et al, Cell Stem Cell 2015. The results obtained with GAL-9 knock-down are encouraging but limited to a single assay, and therefore, seem to be preliminary.

Response: We have significantly improved our discussion of this data based on new experiments conducted in response to this critique.

Using GAL-9 knock-down human B-ALL cells, we have now demonstrated for the first time that GAL-9 is required for the induction of ACM-induced senescence in human B-ALL cells (**Extended Data Fig. 10h**). This data provides a critical mechanistic observation which demonstrates how B-ALL cells evade cell death when exposed to adipocyte-induced genotoxic stress.

Based on additional confocal studies (particularly 3D reconstructions of human B-ALL cells stained for GAL-9 and TIM-3 using Z-stacks), we conclude that while TIM-3 surface expression does not change in response to adipocyte-secreted factors, its intracellular co-localization with GAL-9 increases when leukemia cells are cultured in ACM. Given the recently reported role of TIM-3 trafficking GAL-9 to the surface of acute myeloid leukemia cells, and the increased surface expression of GAL-9 on ACM-cultured human B-ALL cells observed in our study, we posit that TIM-3 is acting as a trafficker of GAL-9 when human B-ALL are in adipose-rich microenvironments.

From GAL-9 knockdown experiments (Extended Data Fig. 10f), we were able to conclude that, 1) knockdown of GAL-9 results in a significant increase in apoptosis of human B-ALL cells only when cultured in ACM, and 2) treating GAL-9 knockdown cells with α GAL-9 antibody no longer induces cytotoxicity. These results demonstrate that GAL-9 is essential for the survival of human B-ALL cells after exposure to adipocyte-secreted factors and confirms the specificity of our GAL-9 targeting antibody for its mode of action. Thus, these data are strongly complementary to data depicted in **Figs. 6, 7, 9 & 10**.

In addition to TIM-3, GAL-9 has several binding partners including CD44, CLEC7a (Dectin-1), and CD137 (41BB). We are actively studying how each of these molecules impact GAL-9-mediated B-ALL pathogenesis. While there are many other molecules that we could query as it relates to GAL-9 function in B-ALL cells, these experiments would do little to strengthen the data already presented demonstrating the identification and validation of GAL-9 as a potential therapeutic target for B-ALL in the context of adiposity, which is the focus of this manuscript. We hope to present our findings regarding other lines of related research in other manuscripts.

2/ "Ultimately, GAL-9 inhibition through of neutralizing antibody in Xenogeneic Model with primary human B-ALL samples will dramatically increases the impact of the study (as previously described for AML by Kikushige et al, Cell Stem Cell 2015)." The authors validate their results on human cell using the 697 human cell line in immunodeficient NOG, which represent a less compelling model than primary human B-ALL

samples, but still a noticeable result.

Response: We appreciate the reviewer recognizing that our xenograft model demonstrating for the first time that GAL-9 blockade protected obese but not lean mice from B-ALL induced mortality is a noticeable result.

Figure 4d: *Confocal representative pictures shown in this figure are not consistent with previously results obtained by flow cytometry analysis. Compared to RPMI condition, Gal9 appears to be upregulated in SCM media in similar way than in ACM media (compared to flow panel in Figure 5a). Tim3 expression seems to be more expressed in ACM than RPMI and SCM conditions (compared to flow panel in Extended Figure 7a). These discrepancies question the robustness of these data and should be explained. The low CD20 expression in the RPMI condition is also surprising.*

Response: The major difference between our confocal and flow cytometry data is that samples processed for confocal experiments are permeabilized; whereas, those processed for flow cytometry assays are not. Therefore, these assays cannot be directly compared because confocal analysis accounts for total GAL-9 levels and our flow cytometry results account for surface-expressed GAL-9.

The confocal analysis indeed show that, even at steady state (RPMI conditions), GAL-9 proteins levels are found in intracellular vesicles (which we confirmed using Z-stack imaging). However, we report that this lectin is redistributed to the cell surface after exposure to adipocyte-secreted factors to slow the proliferation of B-ALL cells attempting to repair damaged DNA.

Confocal analysis also confirms the increased colocalization of GAL-9 and TIM-3 in ACM-cultured human B-ALL cells. Again, our data suggest that TIM-3 is trafficking GAL-9 to the surface of human B-ALL cells after exposure to the adipocyte secretome.

Based on this critique, we have clarified the methodology as well as the interpretation.

We acknowledge the reviewer's critique and have replaced the previous CD20 image in the revised manuscript. We sincerely appreciate the reviewer's careful perusal of our confocal data.

Reviewer 3:
Comments

In this revised manuscript, the authors have made significant revisions to the text and have added a number of additional experiments. Collectively, these changes have dramatically improved the manuscript and all of my points of criticism have been sufficiently addressed.

I would suggest the authors address the following minor points:

1. *Line 124 has a typo and should read "are more resistant".*

Response: We sincerely appreciate the reviewer catching this typo. This error has been corrected. Furthermore, we have attempted to carefully edit our manuscript to eliminate additional errors.

2. *Lines 145-148 describe the nomination of Galectin 9. Beyond "further assessment" what criteria made the authors focus on this molecule?*

Response: After considering this comment, we have included more information in the text which explains why GAL-9 was chosen for further study. This information is presented throughout the text, we have expanded our **Materials and Methods** section, and we have added **Table 1** to address this critique.

3. The text in Figure 4e is too small to be visualized.

Response: We have considerably revamped and simplified **Figure 4**, which now, does not include the original Fig. 4e. We have added a new **Table 1**, which includes a subset of up- and downregulated genes identified in both human B-ALL cells as tracking with methotrexate-induced chemoresistance.

Reviewer 4:
Comments

The study is well-designed, and the results are very interesting and highly relevant to the still obscure relationship between obesity and other diseases, in this case Leukaemia. The main caveat of the RNAseq part of the study is the lack of technical or biological replicates, which the authors partially overcome by qPCR replication of the results. Nevertheless, what the qPCR replication cannot provide are the differentially expressed genes that would be been identified in addition to GAL-9 due to this technical limitation and authors should acknowledge that in the discussion part of the study.

Aside of that, there is some critical information lacking the methodology of RNAseq used on the side of the bioinformatic pipeline which is crucial for evaluation and needs to be included in the manuscript for reproducibility:

1) The authors mention nothing of the alignment to the reads to the reference. In mouse and human data which version was used mm10 and hg38? Something else? Which aligner was used, STAR, bowtie or something else? How did the authors, if in any way, deviated from the best practices/defaults of whatever aligner they used?

Response: We appreciate the reviewer for pointing these omissions. These concerns are now address in the **Materials and Methods** section of our manuscript. In summary, the data were mapped to the hg38 version of the human genome using STAR with the default parameters without deviating from the default settings.

2) What statistical test and, most importantly, multiple testing correction did the authors use for the RNAseq data? FDR?

Response: Gene sets were identified using the multiple change point model (cpm) to successive differences in $-\log_{10}$ transformed sets. For gene expression data, the 'change point' Bioconductor package was used to identify pathways which were significantly changed under each conditioned tested.

3) Which are the 20 candidates for adhesion molecules investigated in the RNAseq data? Which criteria was used for selecting this list and not other? In this test how was multiple testing correction applied?

Response: A Bonferroni correction was used to validate that the significance held true for Galectin-9 (GAL-9), which is our major focus of interest in this manuscript. Additionally, we employed *p*-value thresholds of 0.01 and 0.05 to call significance due to our low sample size. In addition to *LGALS9*, we analyzed the following genes based on their reported signaling and adhesion properties (note cellular aggregation and altered intracellular signaling are hallmarks of adipocyte-exposed human B-ALL cells): *ST3GAL6*, *LGALSL*, *LGALS1*, *LGALS3BP*, *LGALS8*, *CD44*, *CD69*, *ICAM1*, *ICAM2*, *ICAM3*, *ICAM4*, *ICAM5*, *PCDH1*, *PCDHGC3*, *PCDH9*, *CDH24*, *CDH2*, *ITGA4*, and *ITGB1BP1*.

4) The authors state that they removed from further analysis before normalisation genes with no expression or "low expressed" genes? What is the cut-off levels and how many genes did this cut-off?

Response: In the absence of biological replicates per cell line, we considered genes with reasonable coverage (CPM >0.1) in all samples. In all, 13,185 genes met this cut-off. Our data have been uploaded to the GEO repository (GEO accession number GSE183062).

5) How did the authors manage the PCR duplicates in the RNAseq data?

Response: PCR duplicates were marked using the MarkDuplicates function in PICARD. This information is now included in the manuscript. We appreciate the reviewer for making us aware that this information was missing.

6) How did the authors manage the multimapping reads in the RNAseq data?

Response: This is a fantastic question which we apologize for not addressing in the previous submission. Only uniquely mapped reads were used for downstream analysis. We have added this information in the revised manuscript.

We thank you for consideration of our revised manuscript, which we feel is significantly strengthened due to your insightful feedback.

Reviewers' comments:

Reviewer #2 (Remarks to the Author):

The authors have made significant revisions in response to the reviews, and the quality of the manuscript is overall greatly improved. My concerns have been fully addressed. The manuscript now presents a compelling demonstration of the impact of the adipocytic environment on leukemic B cells and the importance of Gal-9 as a novel target to improve outcomes for obese patients with B-ALL.

Reviewer #4 (Remarks to the Author):

The authors have adequately addressed all concerns and recommendations from this reviewer.

We appreciate the reviewers' critiques and the useful experimental suggestions. After addressing the reviewers' concerns, we are happy that our manuscript has been conditionally accepted, pending editorial revisions. Below we have included detailed descriptions of how the reviewers' critiques were addressed, which we feel now makes our manuscript appropriate for publication in *Nature Communications*.

Reviewer 2

Comments: The authors have made significant revisions in response to the reviews, and the quality of the manuscript is overall greatly improved. My concerns have been fully addressed. The manuscript now presents a compelling demonstration of the impact of the adipocytic environment on leukemic B cells and the importance of GAL-9 as a novel target to improve outcomes for obese patients with B-ALL. **Response:** We sincerely appreciate the helpful critiques and comments made by this reviewer, and we are pleased with their endorsement of our work.

Reviewer 4

Comments: The authors have adequately addressed all concerns and recommendations from this reviewer. **Response:** We sincerely appreciate the feedback from Reviewer No. 4 regarding our RNA-sequencing results. We feel that successfully addressing their comments have substantially improved our manuscript.

We thank you for your consideration and insightful feedback of our revised manuscript.